# Neuron-specific chromosomal megadomain organization is adaptive to recent retrotransposon expansions

Sandhya Chandrasekaran [1,2], Sergio Espeso-Gil [1,2], Yong-Hwee Eddie Loh[2,3], Behnam Javidfar[1,2],
Bibi Kassim[1,2], Yueyan Zhu[4], Yuan Zhang[4,5], Yuhao Dong[4], Lucy K. Bicks[1,2], Haixin Li[6,7], Prashanth Rajarajan[1,2],
Cyril J. Peter [1,2], Daijing Sun[4], Esperanza Agullo-Pascual[2], Marina Iskhakova[1,2], Molly Estill[2],
Bluma J. Lesch [6,8], Li Shen [2], Yan Jiang [4,9✉] & Schahram Akbarian [1,2,9✉]

Regulatory mechanisms associated with repeat-rich sequences and chromosomal conformations in mature neurons remain unexplored. Here, we map cell-type specific chromatin domain organization in adult mouse cerebral cortex and report strong enrichment of Endogenous Retrovirus 2 (ERV2) repeat sequences in the neuron-specific heterochromatic $B_2^{NeuN+}$ megabase-scaling subcompartment. Single molecule long-read sequencing and comparative Hi-C chromosomal contact mapping in wild-derived SPRET/EiJ (*Mus spretus*) and laboratory inbred C57BL/6J (*Mus musculus*) reveal neuronal reconfigurations tracking recent ERV2 expansions in the murine germline, with significantly higher $B_2^{NeuN+}$ contact frequencies at sites with ongoing insertions in *Mus musculus*. Neuronal ablation of the retrotransposon silencer Kmt1e/Setdb1 triggers $B_2^{NeuN+}$ disintegration and rewiring with open chromatin domains enriched for cellular stress response genes, along with severe neuroinflammation and proviral assembly with infiltration of dendrites . We conclude that neuronal megabase-scale chromosomal architectures include an evolutionarily adaptive heterochromatic organization which, upon perturbation, results in transcriptional dysregulation and unleashes ERV2 proviruses with strong neuronal tropism.

[1] Department of Psychiatry, Icahn School of Medicine at Mount Sinai, New York, NY 10029, USA. [2] Nash Family Department of Neuroscience, Icahn School of Medicine at Mount Sinai, New York, NY, USA. [3] USC Libraries Bioinformatics Services, University of Southern California, Los Angeles, CA, USA. [4] State Key Laboratory of Medical Neurobiology and MOE Frontiers Center for Brain Science, Institutes of Brain Science, Fudan University, 200032 Shanghai, China. [5] Department of General Surgery, Shanghai Pudong Hospital, Fudan University Pudong Medical Center, 201399 Shanghai, China. [6] Department of Genetics, Yale School of Medicine, New Haven, CT 06520, USA. [7] Clinical Medicine Scientific and Technical Innovation Center, Shanghai Tenth People's Hospital, Tongji University School of Medicine, 200092 Shanghai, China. [8] Yale Cancer Center, Yale School of Medicine, New Haven, CT 06520, USA. [9] These authors jointly supervised this work: Yan Jiang, Schahram Akbarian. ✉email: yan_jiang@fudan.edu.cn; schahram.akbarian@mssm.edu

Repeat-rich sequence blocks, considered major determinants for 3D folding and structural genome organization in the cell nucleus in all higher eukaryotes, are critically involved in a wide range of genomic functions, from lineage-specific gene expression programs in fungi[1] to X-inactivation in early mammalian development[2]. Repetitive DNA may also be important for spatial genome organization in the brain. For example, monogenic neurodegenerative and neurodevelopmental diseases could result from abnormal locus-specific expansion of short-tandem repeats (STR) at the periphery of topologically-associating domains (TADs), a type of conformation defined by chromosomal loop extrusions normally constrained by strong boundary elements at TAD peripheries[3]. However, the relationship between the 3D genome (3DG) and DNA repeat organization in brain cells, including potential implications for neuronal health and function, remains unexplored.

Here, we apply Hi–C, an established method for genome-wide chromosomal conformation mapping including the spatial organization of open ('A-compartment') and closed ('B-compartment') chromatin[4], to show that megabase-scale chromatin domain and compartment organization in adult mouse cerebral cortex is linked, in highly cell type-specific fashion, to multiple retrotransposon superfamilies which comprise the vast majority of 'mobile' DNA elements in the murine genome[5]. Specifically, we identify a neuronal megadomain subtype for which species-specific interaction frequencies track the dramatic reconfiguration of the retrotransposon landscape in *Mus musculus*-derived inbred lines, primarily due to ongoing germline expansions of *Endogenous Retroviruses (ERVs)*, a group of retroelements regulated in highly tissue-specific manner[6], with neuroinflammatory and neurodegenerative potential[7], and detrimental effects on cognition upon de-repression[8]. We show that neuronal deficiency for *Kmt1e/Setdb1* histone methyltransferase, critical to the KMT1E-KAP1-Zinc finger and retrotransposon silencer complex[9,10], triggers massive megabase-scale disintegration and rewiring of chromosomal interactions among chromatin domains anchored in ERV-rich genomic loci. This was associated with retrotransposon un-silencing and severe neuroinflammation and activation of cellular stress genes, intriguingly in close physical proximity to ERV-enriched megadomains, with the endomembrane system of susceptible Setdb1-deficient neurons hijacked for provirus assembly generating provirus-like particles. Our findings provide an example of how 3DG compartmentalization in the mature mouse brain is critically shaped by mobile DNA elements in strictly cell-type fashion, uncovering a distinct heterochromatic regulome in neurons which, upon perturbation, could robustly unleash ERV proviruses.

## Results

### Megadomain organization in adult mouse cerebral cortex.

To study neuronal megadomain organization in adult mouse cerebral cortex, we prepared in situ Hi–C libraries from neuronal (NeuN+), and for comparison, non-neuronal (NeuN−) nuclei from $N = 4$ (2F/2M) mice) and sorted HiC Pro (v2.9) chimeric reads from autosomal sequences by k-means clustering at 250 kb resolution (Fig. 1a, Data S1)[11]. We identified 4 large chromosomal subcompartments in each population (Figs. S1 and S2), reproducible across replicates (Fig. S3), comprising 2.46 Gb of sequence, or >99.9% of the autosomal murine reference genome *mm10*, in NeuN+ (*range*: 104-1460 Mb/subcompartment) and in NeuN− (range: 95–1452 Mb/subcompartment) (Data S2, S3). We assigned identifiers to each first according to minority (A) or majority (B) fractional concordance with heterochromatic, nuclear lamina-associated domain (LAD) sequences, and then numbered based on decreasing size (Fig. 1b). Interestingly, for subcompartments A1, A2, and B1, >60% of loci in NeuN+ and

NeuN− nuclei matched (Fig. 1b). However, within B2, the fractional concordance between neuronal and non-neuronal chromatin dropped to <20% (Fig. 1b), suggesting that in NeuN+, $B_2^{NeuN+}$ interact in a unique manner as compared to NeuN−.

Next, to test whether this neuron-specific subcompartment organization is reproducible across independently generated Hi–C datasets, we applied our k-based subcompartment mapping to three published neuron-specific Hi–C data sources from adult mouse cerebral cortex and hippocampus[12,13], and from neuronal culture differentiated in vitro[14] (Data S4–S6). In fact, $B_2^{NeuN+}$ loci reproducibly clustered together specifically in 3/3 of these previously published neuronal Hi–C datasets (Fig. S4). In striking contrast, Hi–C maps generated from mouse embryonic stem cells (ESC)[14] completely lacked k-clustering of $B_2^{NeuN+}$ sequences (Data S7). Furthermore, *trans* interaction frequencies among $B_2^{NeuN+}$ were consistently high in each dataset generated from mature neurons from adult forebrain, while their surrounding non-neuronal cells ($B_2^{NeuN−}$), like ESC or immature neurons, showed markedly weaker $B_2$ interactions, in contrast to otherwise similar interaction profiles for the remaining subcompartments (Figs. 1c and S4). These findings, taken together, confirm that the neuronal 3D genome includes a cell-type specific B compartment subtype with high inter-chromosomal contact frequencies in mature brain.

Of note, chromatin profiling (mouseENCODE[15]) identified two subcompartments in NeuN+ ($A_1^{NeuN+}$ and $A_2^{NeuN+}$) and NeuN− ($A_1^{NeuN−}$ and $A_2^{NeuN−}$) with high fractional concordances for enhancers and chromatin marks broadly associated with actual or potential transcription (Fig. 1d). Uniquely, $B_2^{NeuN+}$ revealed a high (60.1%) fractional concordance with the H3K9me3 repressive histone mark, unseen in $B_2^{NeuN−}$ (Fig. 1d). In fact, neuronal $B_2^{NeuN+}$ megadomain sequences showed a highly specific, significant association ($p = 3.36 \times 10^{-41}$, Fisher's $2 \times 2$ exact test) with adult cortex NeuN+ H3K9me3, but not with active histone marks (Fig. 1e, f and Table S1)[16], also unseen in NeuN− nuclei. In addition, we noticed that 89.6 Mb (86.2%) of $B_2^{NeuN+}$ sequences anchored most neuron-specific inter-autosomal contacts ($p < 10^{-50}$, Homer v4.8, binomial testing, 1 Mb resolution) were not shared with non-neuronal cells (Fig. S5 and Data S8, S9). As a result, *trans* contact maps diverged by cell type, with striking consistency across the 4 animals (Pearson's r range: NeuN+ [0.81–0.95] and NeuN− [0.95–0.99]) (Fig. 1g). Interestingly, neuron-specific *trans* interactions predominantly involved $B_2^{NeuN+}$ loci, while non-neuron-specific *trans* interactions primarily involved $A_1^{NeuN−}$ and $A_2^{NeuN−}$ (Figs. S4 and S6), with sharply divergent, highly cell-type specific trans-contact profiles as illustrated in the 50 Mb wide browser windows from several chromosomes (Fig. 1h and Figs. S6 and S7). Importantly, loci comprising $B_2^{NeuN+}$ interacted significantly more frequently within NeuN+ as compared to NeuN− nuclei (Fig. S8). Notably, several gene clusters, including protocadherins, vomeronasal receptors, and cytochrome p450 enzymes, are encompassed within $B_2^{NeuN+}$ megadomain interactions (Fig. S9). Robust interchromosomal Hi–C contacts of olfactory receptor gene clusters were previously reported in olfactory sensory neurons[17].

### Neuron-specific megadomain organization tracks recent expansions in ERV genomic landscapes.

Transposable elements, simple repeats and other types of repetitive DNA comprise up to 45% of the mouse genome[18]. Therefore, we asked whether our megadomain subcompartments show repeat-specific enrichment. We screened our four subcompartments against 10 broad classes of repeat categories (Data S10). Indeed, $B_2^{NeuN+}$ showed a significant ($p = 2.13 \times 10^{-35}$, Fisher's $2 \times 2$ exact test) enrichment for hotspots (99th percentile densities) of the *Endogenous Retrovirus* subtype

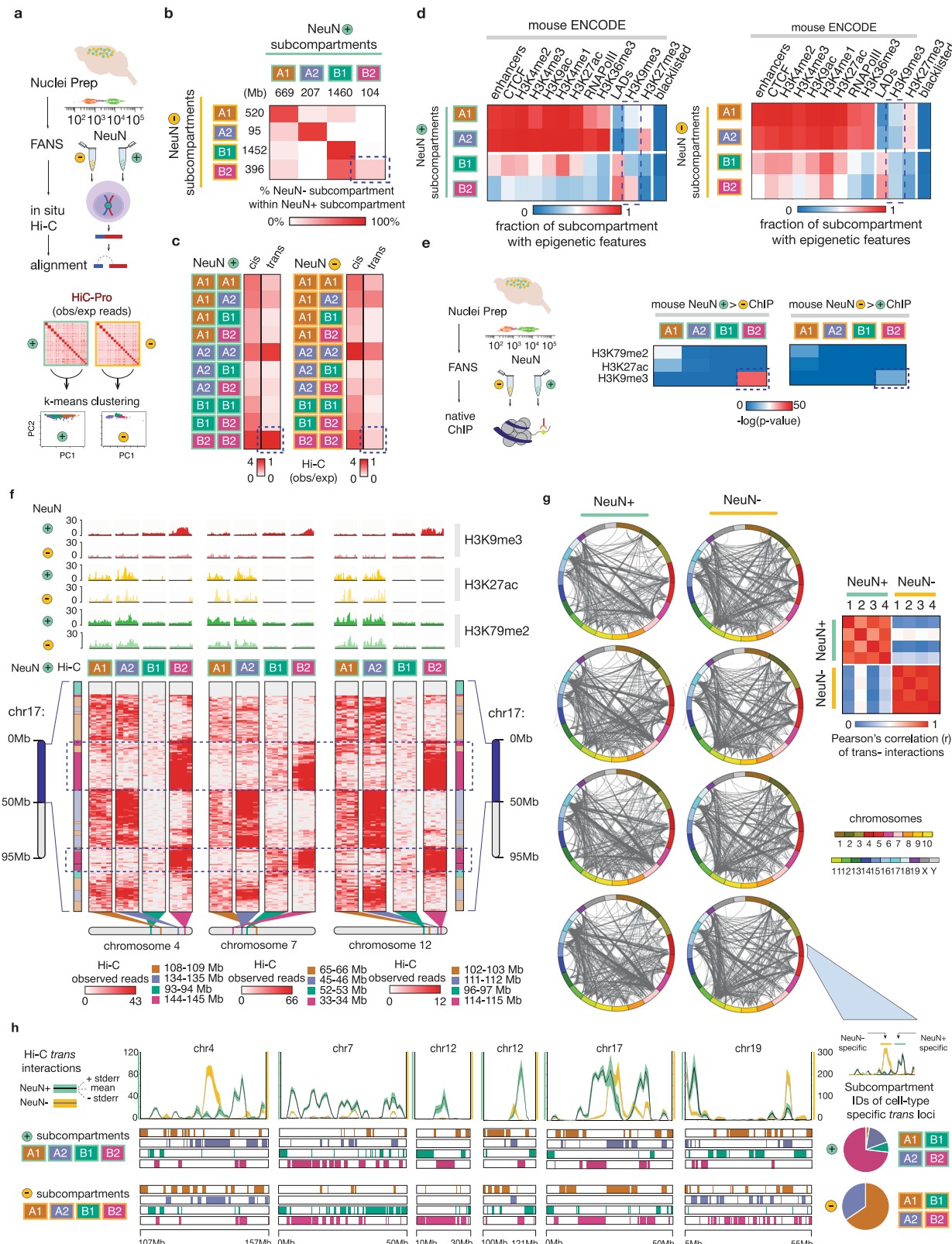

*ERV2* (Fig. 2a). The *ERVs* are a vertebrate-specific RNA-based ('copy-and-paste') retrotransposon family defined by long-terminal repeats (LTRs) flanking the core *gag-pol-env* viral coding domains[19,20]; as a class, ERVs colonize ~10% of the mouse genome (ERV2 specifically comprises 3.14%)[21]. Remarkably, none of the remaining neuronal subcompartments and none of the non-neuronal subcompartments showed *ERV2* enrichment. However,

both $A_2^{NeuN+}$ and $A_2^{NeuN-}$ showed significant enrichment ($p = 2.47 \times 10^{-55}$ (NeuN$^+$), $p = 3.11 \times 10^{-42}$ (NeuN$^-$), Fisher's $2 \times 2$ exact test) for *SINE* (*Short-Interspersed Nucleotide Elements*) non-autonomous retroelements[19] previously linked to domain boundaries and inter-chromosomal interactions in some metazoan systems[22–24], serving as CTCF binding scaffolds (Fig. S10). The $A_1$ subcompartment in both cell types was significantly enriched for the

**Fig. 1 Cell-type specific subcompartment architectures in adult mouse cerebral cortex. a** (Left) Workflow. Fluorescence activated nuclei sorting (FANS) was performed on dissected adult mouse cortical tissue ($n = 4$, 2 F/2 M), followed by in situ Hi-C on intact NeuN+ and NeuN− nuclei. Genome alignment (HiC-Pro (v2.9)) was facilitated by split-mapping at the known chimeric ligation junctions to generate genome-wide pairwise contact maps. Hi-C read counts for NeuN+ and NeuN− (observed/expected) were then independently piped into a k-means clustering algorithm, ultimately generating four clusters representative of chromatin subcompartments in each population. **b** Correspondence of the NeuN+ and NeuN− determined subcompartments; genome extents of each subcompartment as indicated. Percent overlap of coordinates from each NeuN− subcompartment with each of the NeuN+ subcompartments are represented on the indicated color scale of white (0%) to red (100%). **c** Heatmap of mean Hi–C (observed/expected) read counts for NeuN + (left) and NeuN− (right) between loci comprising the designated subcompartments; 250 kb resolution. **d** Heatmap summarizing the characterization of determined clusters as 'A' (active) or 'B' (inactive) with lamin-associated domains (LADs), enhancer coordinates, and mENCODE ChIP-Seq data for RNAPolII, CTCF, several histone modifications, and mENCODE blacklisted sequences in NeuN+ (left) and NeuN− (right), as indicated. The fraction of 250 kb bins comprising the cluster overlapping these genome tracks is indicated on a scale of 0 (blue) to 1 (red). **e** Enrichment heatmap of differential NeuN+ and NeuN− H3K79me2 ($n = 2$), H3K27ac ($n = 3$), and H3K9me3 ($n = 4$) tagged sequences with loci comprising each of the four subcompartments (Fisher's $2 \times 2$ exact testing, one-sided) (Left) NeuN+ histone modification enrichment, diffReps ($p < 0.001$). (Right) NeuN− histone modification enrichment, diffReps ($p < 0.001$). **f** Representative NeuN+ long-range contact patterns (250 kb resolution) for multiple genomic intervals comprising each chromatin subcompartment. Normalized NeuN+ and NeuN− histone modification profiles tracks of NeuN+ and NeuN− H3K79me2 ($n = 2$), H3K27ac ($n = 3$), and H3K9me3 ($n = 4$) are included. **g** (Left) Individual Circos plots from four biological replicates for NeuN + (left) and NeuN− (right) trans-chromosomal contacts (1 Mb resolution, HOMER (v4.8), threshold for significant interactions displayed (binomial testing): $p < 10^{-50}$; see Methods section). NeuN+ and NeuN− Circos plots aligned horizontally originate from the same brain tissue. Zoom-in window bottom right show 50 Mb portion of chr4; notice sharply different trans profiles for NeuN+ and NeuN−. (Right) Pearson's correlation heatmap of loci involved in trans- interactions in NeuN+ and NeuN− across replicates ($n = 4,4$). Scale as shown ($r = 0$ (blue), $r = 0.5$ (white), $r = 1$ (red)); 1 Mb resolution. **h** (Left) Hi-C trans contact maps for NeuN+ (green) and NeuN− (orange) across 50 Mb windows from six chromosomes, as indicated. NeuN + ($n = 4$) mean trans interactions in black, SEM in green; NeuN− ($n = 4$) mean trans interactions in gray, SEM in orange. Subcompartment designations in NeuN + (top) and NeuN− (bottom) included. Y-axis shows number of significant trans contacts (HOMER, threshold: $p < 10^{-50}$) (green) for NeuN+ and (yellow) NeuN− nuclei. Source data are provided as a Source Data file.

largely fossilized ERV3 TEs ($p = 6.34 \times 10^{-26}$ (NeuN+), $p = 6.39 \times 10^{-15}$ (NeuN−), Fisher's $2 \times 2$ exact test). Furthermore, LINE (Long-interspersed Nucleotide Element) TEs showed strong enrichment for the B1 compartment for both cell types ($p = 2.37 \times 10^{-10}$ (NeuN+), $p = 5.26 \times 10^{-8}$ (NeuN−), Fisher's $2 \times 2$ exact test). Thus, each major type of TE shows unique, subcompartment-specific enrichment patterns in brain, with neuron-specific $B_2^{NeuN+}$ megadomains tagged with high levels of H3K9me3 and enriched for ERV2 elements.

Of note, a subset of ERVs with preserved retrotransposition capacity, including intracisternal A particles (IAPs), continue to invade the murine germline, accelerating their expansion within *Mus musculus*-derived inbred lines[25]. As a result, IAPs comprise >80% of all ERVs genome-wide in *C57BL/6J* laboratory inbred mice, in stark contrast to the very low (<10%) phylogenetic conservation of those IAPs in the genomes of wild-derived *SPRET/EiJ* mice[25]. We wondered whether inter-chromosomal contacts, comprising the defining feature of the $B_2^{NeuN+}$ subcompartment, occur in reduced frequencies in *SPRET/EiJ* vs. *C57BL/6J* neurons. To explore, we prepared Hi-C libraries from sorted adult cortex NeuN+ and NeuN− nuclei of both mouse strains in parallel ($n = 2$ (1 F/1 M/strain)) (Fig. 2b). Indeed, hotspots of ERV2 repeats ($n = 101$ 250 kb bins), defined as >99th percentile densities of ERV2 in the *mm10* reference genome, interacted less frequently in *SPRET/EiJ* versus *C57BL/6J* NeuN+ ($p = 2.25 \times 10^{-18}$, Wilcoxon sum-rank testing); in comparison, interaction frequencies at these loci minimally differed in NeuN− nuclei across the two strains ($p = 0.001$, Wilcoxon sum-rank testing) (Fig. 2c), supporting a role for ERV2 densities in shaping the 3D NeuN+ genome. Furthermore, overall intra- and inter-chromosomal connectivity across the 4 neuronal subcompartments showed minimal changes between *SPRET/EiJ* and *C57BL/6J* (Fig. S11). We conclude that $B_2^{NeuN+}$ megadomain conformations in mature cortical neurons show 'dosage-sensitivity' for phylogenetically young ERV2 sequences.

We suspected that many IAP integration sites in our profiled mouse brains are, in fact, not represented in the reference genome due to a variety of factors including residual genetic contribution

from non-C57BL/6J lines, genetic drift, or somatic retrotransposition in stem cells and progenitor cells during early embryogenesis or in the developing brain. To explore whether the observed IAP/ERV2 enrichment of the $B_2^{NeuN+}$ subcompartment is preserved beyond the retroelement sequences annotated in *mm10*, we used biotinylated oligonucleotides to capture 10-15 kb sized DNA fragments carrying phylogenetically young and retrotransposition-competent IAP subtype *IAPEzi* (6481 bp)[6,21,26] together with surrounding flanking sequences for genomic annotation (Fig. 3a, b). We ran single molecule PacBio SMRT-sequencing (SMRT-seq) on captured DNA of sorted NeuN+ neuronal and, separately, NeuN− non-neuronal nuclei collected from adult cerebral cortex from two independent mouse colonies, generating $2.64–4.70 \times 10^6$ high fidelity (HiFi, >99.9% accuracy) circular consensus sequences (CCS) for a total of 8 cell-type specific samples (5 NeuN + , 3 NeuN−) passing QC metrics and sent for next-generation sequencing (Tables S2 and S3, Data S11, see Methods section). For each sample, the overwhelming majority of *IAPEzi* sequences (83–94%) expectedly annotated to *IAPEzi GRCm38/mm10* sites (Repeatmasker). However, each of the eight profiled samples identified 209–684 proviral (non-solo-LTR) *IAPEzi* integration sites not found in *mm10*, including 2–44 sites/animal harboring full-length *IAPEzi*. Strikingly, proviral (non-solo LTR) sequences, especially full-length *non-mm10 IAPEzi*, showed a significant 2.5- to 4-fold enrichment for $B_2^{NeuN+}$ sequences compared to the remaining subcompartments ($p = 0.0391$, paired Wilcoxon sum-rank testing, $B_2^{all}$ vs. $B_2^{full}$) (Figs. 3c, d and S12, and Data S12). We conclude that full-length, potentially retrotransposition-competent IAP elements continue to preferentially insert into genomic loci 'destined' to assemble as $B_2^{NeuN+}$ subcompartment.

Importantly, the number of IAPEzi genome integration sites in cortical NeuN+ neurons was only minimally different from the corresponding counts in the non-neuronal (NeuN−) nuclei (Fig. 3c and Data S12). To examine this further, we conducted additional SMRT-seq of IAPEzi integration sites from two samples of male germ cells, prepared as post-meiotic round spermatid cultures. These experiments revealed an additional set

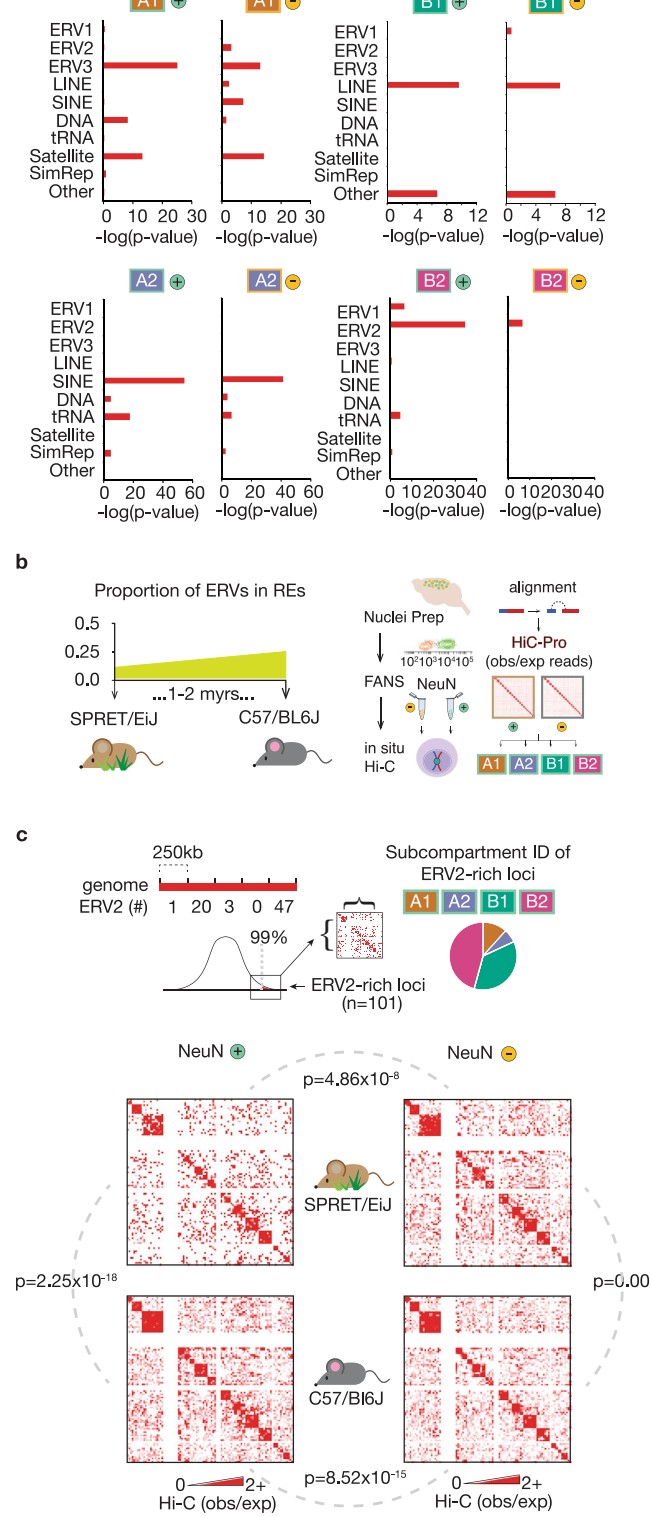

**Fig. 2 Phylogenetic expansion of ERV2 repeat elements impact megadomain organization in neuronal nuclei. a** Repetitive element (RE) associations with loci comprising NeuN + (left) and NeuN− (right) subcompartments. REs (Repbase) classified as indicated. −Log(p-values) displayed as red bars (Fisher's 2 × 2 test, one-tailed). **b** (Left) Proportion of ERVs within REs in SPRET/EiJ and C57BL/6J; adapted from Nellaker et al.[25]. (Right) FANS on dissected adult SPRET/EiJ and C57BL/6J mouse cortical tissue ($n = 4$, 2 M/2F), followed by in situ Hi-C on the sorted intact NeuN+ and NeuN− nuclei, aligned to the genome using HiC-Pro (v2.9) to generate Hi–C pairwise interaction matrices. **c** (top) Schematic of binning and (bottom) summary matrix of NeuN+ (left) and NeuN− (right) Hi–C values (observed//expected) denoting interaction frequencies between genomic loci harboring high densities of ERV2 (>99th percentile, 250 kb resolution) in SPRET/EiJ vs. C57/BL6J mice. Each matrix represents a sum of Hi-C (obs/exp) comprising two biological replicates. Statistical significance of differences as indicated (Wilcoxon sum-rank test, two-sided, paired): SPRET/EiJ NeuN+ vs. C57BL/6J NeuN + ($p = 2.25 \times 10^{-18}$); SPRET/EiJ NeuN− vs. C57BL/6J NeuN− ($p = 0.001$); SPRET/EiJ NeuN+ vs. SPRET/EiJ NeuN− ($p = 4.86 \times 10^{-8}$); and C57BL/6J NeuN+ vs. C57BL/6J NeuN− ($p = 8.52 \times 10^{-15}$). Source data are provided as a Source Data file.

**Neuronal megadomain reorganization upon ablation of Setdb1 methyltransferase.** Next, we generated *CamK-Cre+,Setdb1[2lox/2lox]* conditional mutant mice for neuron-specific ablation of the *Kmt1e/Setdb1* histone methyltransferase, an essential regulator for ERV repression in stem cells and somatic tissues[9] and part of a repressor complex with SMARCAD1 chromatin remodeler and KRAB-associated protein 1 (KAP1)-KRAB zinc-finger proteins[10]. Visual inspection of histological sections from mutant and wild-type adult cerebral cortex processed by DNA-FISH with a 300 bp IAPEzi gag probe and counterstained with NeuN immuno-fluorescence and the nucleophilic dye, DAPI, revealed patch-like accumulations of ERV/IAP-containing genomic sites, with some the largest densities most prominently visible in wildtype NeuN+ nuclei (Fig. S13), consistent the high physical interaction frequencies of ERV-rich chromosomal loci in Hi–C. Therefore, in order to test whether *Setdb1* could exert a broad regulatory footprint of *Setdb1* on ERV-rich megadomains, including compartmental organization and chromosomal connectivity of neuronal $B_2^{NeuN+}$, we generated two NeuN+ libraries to supplement the two previously published cell-type Hi–C chromosomal contacts maps from adult *CamK-Cre+,Setdb1[2lox/2lox]* with littermate controls (CamK-Cre−, Setdb1[2lox/2lox]) ($N = 4$/group (2M/2F)[27] (Figs. 4a and S14). Differential analysis by genotype revealed widespread megadomain rewiring in mutant neurons -- $n = 446$ significant decreases in pairwise *trans* interactions, the vast majority (60.76%) of which reflected loss of $B_2$–$B_2$ *trans*, and another $n = 196$ significant pairwise increases, of which 72.45% represented gains in $B_1^{NeuN+}$–$B_2^{NeuN+}$ *trans* ($p_{adj} < 0.05$, DESeq2 negative binomial testing, 1 Mb bins) (Fig. 4b, *Data S13, S14*). Furthermore, mutant neurons showed significant shifts in intra-chromosomal *cis* Hi–C interactomes, overwhelmingly driven by ERV-enriched subcompartment sequences (42.4% of $N = 4066$ representing involving $B_2^{NeuN+}$ contacts). This included significant losses in cross-compartmental $A_2^{NeuN+}$–$B_2^{NeuN+}$ contacts ($N = 3.79$ cis contacts/per 10 Mb $A_2^{NeuN+}$), representing a 2-fold enrichment as compared to $A_1^{NeuN+}$-$B_2^{NeuN+}$ contacts ($N = 2.05$ cis contacts/per 10 Mb $A_1^{NeuN+}$) (Fig. 4b and Data S13, S14). Of note, this biased disintegration of $A_2^{NeuN+}$–$B_2^{NeuN+}$ *cis* contacts after neuronal *Setdb1* ablation was highly specific because only 0.4 $A_2^{NeuN+}$–$B_2^{NeuN+}$ cis contacts/10 Mb $A_2^{NeuN+}$ showed mutant-specific gain, while the number of $A_1^{NeuN+}$–$B_2^{NeuN+}$ contacts gained was 3.01 cis contacts/10 Mb $A_1^{NeuN+}$.

of 439–488 proviral (non-solo-LTR) IAPEzi integration sites not found in mm10, including 47–59 sites/culture harboring full-length IAPEzi. Strikingly, de novo integration sites from these germ cell samples, compared to IAPEzi SMRT-seq integration sites identified in brain cells, showed a very similar type of enrichment for sequences assembling as $B_2^{NeuN+}$ (Fig. 3c, d). These findings, taken together, effectively rule out neuron-specific retrotransposition as a driver for the observed cell type-specific chromosomal interactions in neuronal nuclei, including $B_2^{NeuN+}$.

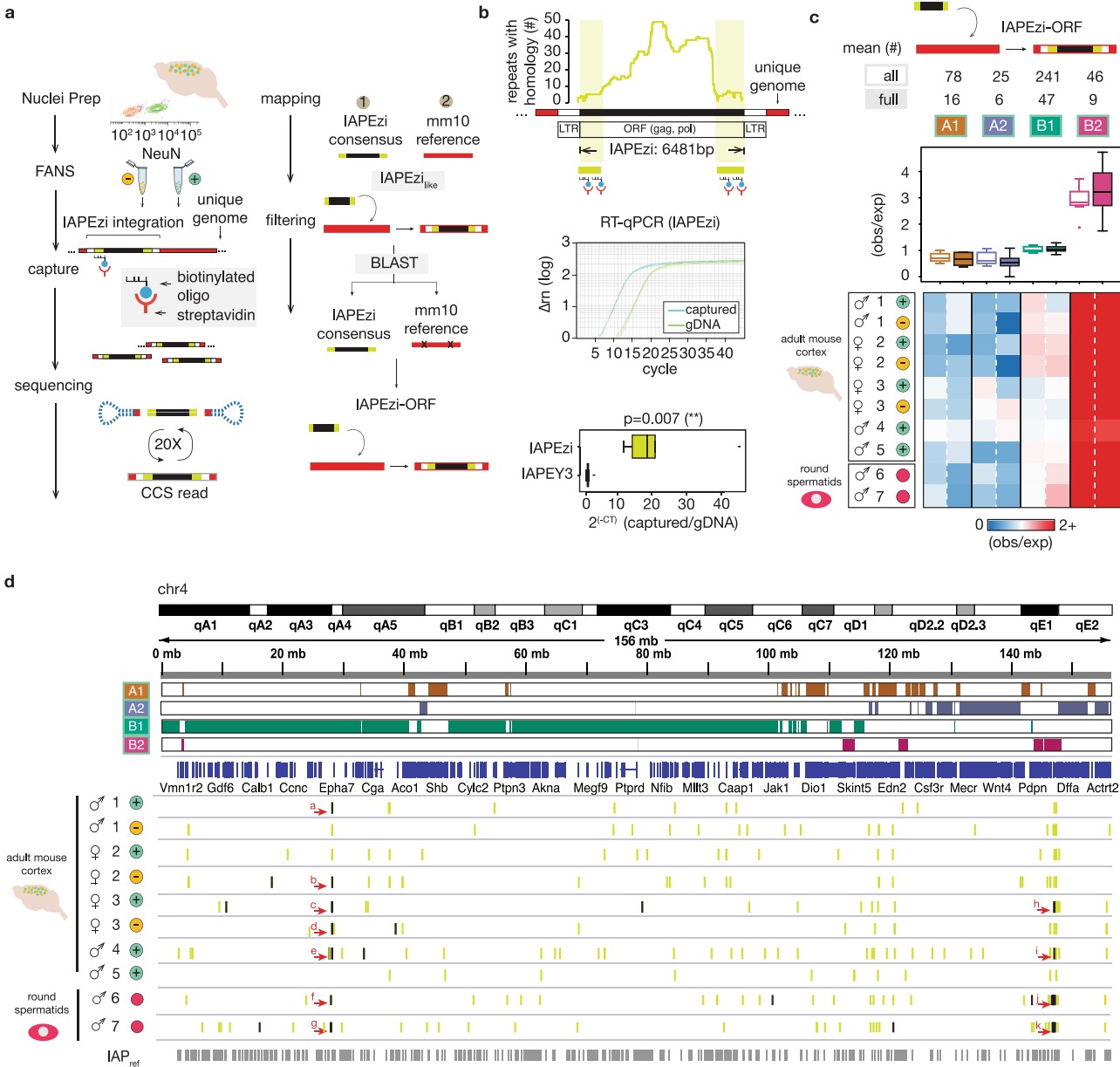

**Fig. 3 Cell-type specific long read sequencing of a mobile ERV2 element. a** Schematic of PacBio SMRT circular consensus sequence (CCS) long-read sequencing workflow. De novo reads identified by evaluating BLAST bit scores between alignment to the IAPEzi consensus and the IAP-masked reference genome. **b** (Top) PacBio biotinylated oligonucleotide probe design. The probe sequences are the most divergent regions of IAPEzi across other repetitive elements with detectable homology (<10%, *dfam.org*). (Middle) Real time qPCR plot of genomic DNA vs. IAPEzi-captured DNA. (Bottom) Box and whisker plot documenting significant IAPEzi enrichment as compared to fossilized IAPEY3 repeat element ($n = 6$ biologically independent samples) ($p = 0.008$, Student's $t$ test, two-sided). **c** Subcompartment box-and-whiskers observed/expected plot (top) and mean observed/expected heatmap for IAPEzi de novo integration sites (bottom). Data shown for eight mouse cortical NeuN+ and NeuN− samples, and two round spermatid samples (totaling $n = 10$) from $n = 7$ biologically independent samples, as indicated. 'Full' de novo reads refer to de novo reads with >90% (5833/6481 bp) match with the IAPEzi consensus sequence length (*dfam.org*). Filled boxes refer to 'all' de novo, while outlined boxes refer to 'full' de novo. **d** Genome browser shot of chr4 with de novo IAPEzi insertions for each of 10 profiled adult cortex neuronal and non-neuronal samples and round spermatids, as indicated. Subcompartment designations as indicated. Arrows point to representative full-length de novo insertions in close genomic range of each other across different samples. Specific coordinates corresponding to all (green) vs. full (black) de novo in *Data S12* Letters refer to de novo insertions labeled in *Data S12* for reference. Source data, included mean ± SEM for the included boxplots, are provided as a Source Data file.

Importantly, both *trans* and *cis* Hi–C alterations upon *Setdb1* ablation were significantly associated with ERV2 hotspots (*trans* losses, $p = 2.95 \times 10^{-30}$; *cis* gains, $p = 2.62 \times 10^{-27}$; *cis* losses, $p = 2.63 \times 10^{-23}$; Fisher's $2 \times 2$ exact testing) (Fig. 4c). To examine whether these changes in compartment-specific chromosomal conformations are associated with transposon un-silencing, we RNA-seq profiled the cortical transcriptome of adult mutant and control mice ($N = 6$/group) (Table S4). Importantly, comparison of subcompartment-specific expression of ERV2s and other repeat classes in mutant and control cortex revealed that increased ERV2 expression in *Setdb1*-deficient mice primarily originated within $B_2^{\text{NeuN+}}$, while the remaining subcompartments showed minimal alterations in transposon expression in comparison to control animals (Fig. 4d).

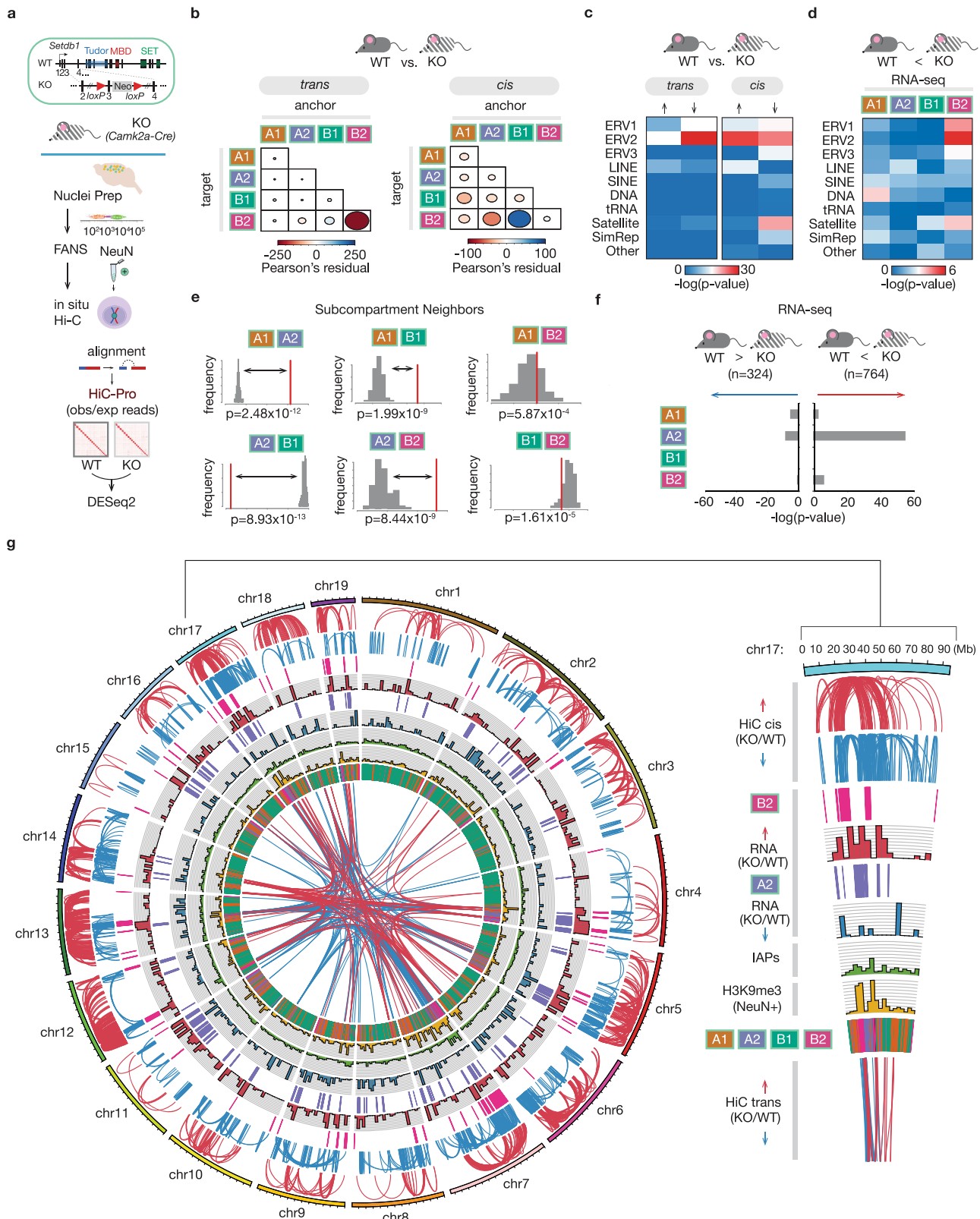

Furthermore, expression of the 764 unique autosomal gene transcripts significantly up-regulated in mutant cortex ($p_{adj} <$ 0.05, DESeq2 negative binomial testing) largely originated in $A_2^{NeuN+}$ ($p = 3.56 \times 10^{-55}$, Fisher's $2 \times 2$ test), the A subcompartment that not only preferentially lost *cis* contacts with $B_2^{NeuN+}$ (as mentioned prior), but also most frequently neighbors $B_2^{NeuN+}$ genome-wide (Fig. 4e, f). These strong, megadomain-

specific biases were not mirrored in the fraction of 324 unique gene transcripts significantly down-regulated in *Setdb1*-deficient cortex (Fig. 4f). Of note, of the 764 unique upregulated autosomal gene transcripts, only a very small subset of 46 genes (6%), of which 36 were located in $B_2^{NeuN+}$, had shown significant NeuN+ specific H3K9me3 enrichment at baseline ($p < 0.001$, *diffReps* negative binomial testing) (Fig. S15). These findings, taken

**Fig. 4 Neuronal megadomain dysregulation after Setdb1 ablation. a** Workflow. (Top) Representation of conditional Setdb1 ablation (KO), adapted from Jiang et al.[27]. (Bottom) FANS was performed on dissected adult KO mouse cortical tissue ($n = 4$, 2F/2M), followed by in situ Hi-C on intact NeuN+ nuclei. Genome alignment (HiC-Pro (v2.9)) facilitated by split-mapping at the known chimeric ligation junctions to generate genome-wide pairwise contact maps. **b** Differential Hi-C trans (left) and cis (right) interactions in KO vs. WT ($n = 4$ (2F/2M)/genotype) ($p_{adj} < 0.05$) (1 Mb resolution). The difference in Pearson's residuals (observed/expected) (r) between the distribution of increased interactions and decreased interactions, with the area of each ellipse representing $r_{absolute}$ (|r|), and color of each ellipse representing r. **c** Genome-wide associations of repetitive element families with loci involved in the altered trans and cis interactions in KO vs. WT. **d** Genome-wide associations of differentially increased RNA-Seq ($n = 6$ (3F/3M)/genotype) of repetitive element families with subcompartment loci. In **c**, **d** $-\log(p\text{-values})$ represent outcomes from Fisher's 2 × 2 testing (one-tailed). **e** Proximity analyses of subcompartment loci in cis. Relative frequencies of contiguous subcompartment block neighbors in the reference genome displayed in red, while the distribution of 100 permutations (regionER, resampleRegions, two-sided) in a random genome with equivalent proportions of such blocks is displayed in gray. **f** Genome-wide associations of differential (both increased and decreased) RNA-Seq ($n = 6$ (3F/3M)/genotype) transcripts by subcompartment ($-\log(p\text{-values})$, Fisher's 2 × 2 testing (one-tailed)). **g** (Left) Circos plot representation of multiple epigenomic features in KO vs. WT NeuN+. (Right) Pie slice displays chr17 with associated legend for individual tracks applicable to the entire Circos plot, including (to top) HiC trans interaction changes (KO vs. WT), H3K9me3 enrichment, IAP densities, RNA changes (KO vs. WT), and HiC cis interactions changes (KO vs. WT). Note the concordance of the B2 subcompartment with H3K9me3 enrichment and foci among the Hi-C tracks, both in *cis* and *trans*; similarly, note concordance of the A2 subcompartment with upregulated transcriptional changes (red) in KO/WT. Source data are provided as a Source Data file.

together, strongly suggest that genes with increased expression after *Setdb1* ablation are primarily affected by loss of chromosomal interactions with repressive $B_2^{NeuN+}$ chromatin but are unlikely to be regulated by Setdb1 at the site of the gene body. Importantly, despite the spatial colocalization of the increased genic transcription in $A_2^{NeuN+}$ and the known ERV2 hotspots in $B_2^{NeuN+}$, we observed no fusion transcripts in any of the profiled RNA-seq samples (Table S5). Together, these findings strongly suggest that *Setdb1*-dependent epigenomic regulation of neuronal megadomains critically regulates $A_2^{NeuN+}$ and $B_2^{NeuN+}$ on a genome-wide scale (Fig. 4g).

**Gliosis and genomic activation of microglia associated with IAP invasion of neuronal somata and processes.** We next asked whether the disintegration of $B_2^{NeuN+}$ chromosomal connectivity, including ERV2 de-repression, in our *Setdb1*-deficient mice could affect neuronal health and function. To this end, analysis of the RNA-seq from *Setdb1* mutant as compared to control cortex revealed that the top 10 ranking gene ontology groups of up-regulated genes included regulators of ribosomal protein synthesis, the endoplasmic reticulum/endomembrane (ER/EM) stress response, ATP-dependent metabolism and the complement cascade (Figure S16, *Data S15*), potentially indicating a hypermetabolic state in response to an inflammatory stimulus. Strikingly, the cerebral cortex and striatal areas of *CamK-Cre+,Setdb1^{2lox/2lox}* conditional mutant mice were affected by gliosis and exhibited upregulation of the astrocytic marker, glial fibrillary acid protein (GFAP) (Fig. 5a, b). Similarly, Iba1 immunostaining marker revealed proliferative and reactive microglia in mutant hippocampus, although labeling of the cortex overlying the hippocampus was not significantly different from control (Fig. 5a, c, d). Nonetheless, given the central role of microglia in brain inflammation, we conducted cell-type specific open chromatin (Fig. 5e) and RNA-seq transcriptome (Fig. 5f) profiling on immuno-panned microglia extracted from adult mutant forebrain of neuronal *Setdb1*-deficiency and controls ($N = 3$/group). Expression of *Setdb1* transcript, including the loxP flanked exon 6 subjected to neuron-specific deletion in the mutant cortex, was completely preserved in the microglia from *CamK-Cre+,Setdb1^{2lox/2lox}* mice (Fig. S17). Furthermore, the overwhelming majority of retrotransposon transcripts, including the entire set of ERV2s, did not show elevated expression in microglia from mutant cortex (Data S16); these findings were expected given that *Setdb1* ablation is restricted to neurons in our conditional mouse model. We identified 840 (629 up, and 211 down) differentially regulated microglia-specific transcripts after neuronal *Setdb1* ablation (Fig. 5f, Data S17). Among these were

629 up-regulated transcripts, for which gene ontology analyses indicated robust activation of interferon and cytokine signaling pathways and blood vessel formation including many genes associated with autoimmune and neuroinflammatory disease (Fig. 5f, Data S17)[8]. In contrast, no pathway enrichment was observed for the group of 211 microglial genes downregulated in mutant cortex. Next, given that the microglial genomic response to inflammatory stimuli also involves widespread changes in chromatin accessibility[28], we profiled open chromatin landscapes on a genome-wide scale using Assay for Transposase Accessible Chromatin (ATAC-seq) on CD11b-immunopanned microglia from mutant and control forebrain ($N = 3$/group) (Fig. 5e, Data S18). We identified 4154 open chromatin regions differentially regulated between two groups. Strikingly, microglial open chromatin region (OCR) upregulated after neuronal Setdb1 ablation showed the strongest enrichment ($p < 10^{-43}$, HOMER binomial testing) for binding motifs of Signal transducer and activator of transcription 2 (Stat2), a regulator of gene expression highly sensitive to activation by interferon and antiviral response pathways in brain and other tissues[29,30]. Furthermore, nuclear factor κB (NF-κB), a key factor in mediating the microglial genomic response to neuronal injury and inflammation[31], was among the top 5 ranking motifs in OCR upregulated in microglia from cortex with neuronal Setdb1-deficiency (Fig. 5e). These findings, taken together, suggest that neuron-specific epigenomic un-silencing of ERV retrotransposons could trigger an inflammatory response with astrocytosis and microglial activation.

Studies in mice with severe immunodeficiencies and in genetically engineered cell lines suggest that the un-silencing of ERVs triggers an immune response primarily via RNA-sensing associated with the mitochondrial antiviral signaling protein (MAVS) and the Stimulator of Interferon Genes (STING) signaling pathways[8,32]. We observed that in our *CamK-Cre+, Setdb1^{2lox/2lox}* mice with neuronal Setdb1 depleted, there was increased transcription specifically of ERV2s, in conjunction with significantly decreased H3K9me3 levels at ERV2 sequences in neuronal chromatin ($p = 0.02366$, Linear regression) (Figs. 6a and S18). In contrast, non-neuronal chromatin from *CamK-Cre+,Setdb1^{2lox/2lox}* mice showed complete preservation of ERV2-bound H3K9me3 (Fig. S19 and Data S19, S20). In addition to these RNA-seq based studies, we also observed increased transcription of IAP-gag in cortical neurons by RNA FISH (Fig. 6b). To assess the viral burden in our mouse model, we next monitored protein expression and found that mutant, but not control cortex showed robust neuronal expression of the IAP Gag protein critical for retroviral assembly (Fig. 6c, d). Electron microscopy confirmed dramatically increased numbers of mutant

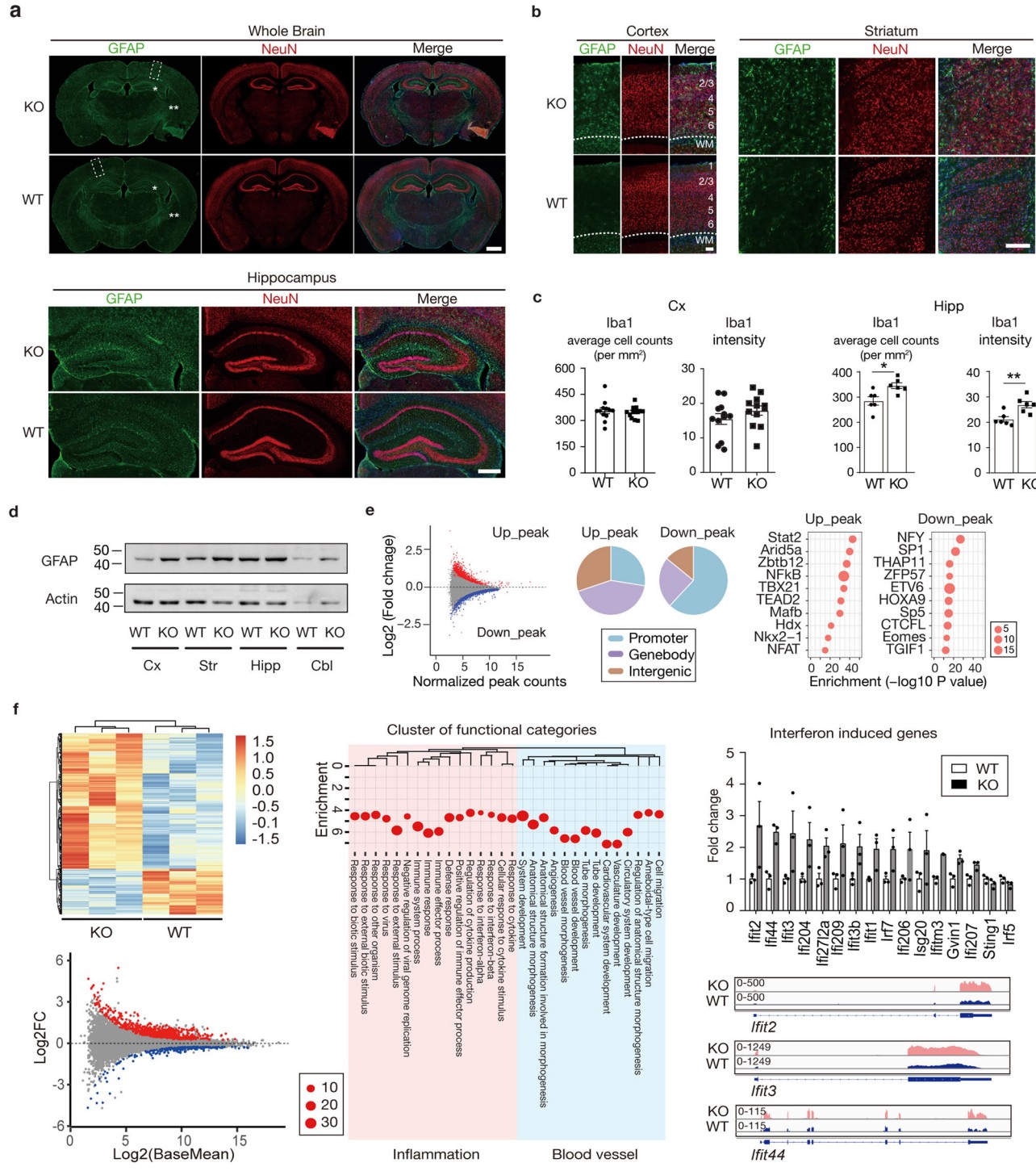

neurons with the presence of immature and mature IAP provirus in close proximity to cisterna-like membranous spaces compared to controls ($p = 3.69 \times 10^{-5}$, Student's two-sided $t$-test). In addition, counts of IAP particles (within the subset of IAP + neurons in both genotypes) were much higher in mutants compared to controls ($p = 0.008$, Wilcoxon sum-rank testing) (Fig. 6e). Strikingly, a subset of neurons showed dramatic proviral proliferation with encroachment into neuronal somata, dendrites, and even spines in a subset of cortical neurons (Fig. 6f). We conclude that the rewiring and epigenomic reorganization of ERV2-enriched neuronal megadomain sequences upon *Setdb1* ablation results in IAP retrotransposon escape from silencing. In addition to the robust increase in IAP transcript and protein

levels, we observed proviral assembly and potentially dramatic accumulation of such particles in susceptible neurons resulting in viral hijacking of endomembrane systems, an ER/EM stress response, and ultimately neuroinflammation.

## Discussion

We report that megadomain organization in the adult brain involves a unique, neuron-specific signature, including a B-type subcompartment encompassing 104 Mb of neuronal chromatin ($B_2^{NeuN+}$) composed of comparatively 'small' (1–12 Mb) heterochromatic 'islands' engulfed by $A_2^{Neuron}$ and other A-compartment-associated megadomains. Comparison of cell-type

**Fig. 5 Astrocytosis and microglial activation in Setdb1-deficient forebrain. a** Representative images show immunofluorescent signal for GFAP (green) and NeuN (red) in (top) whole brain coronal section (top) and (bottom) hippocampal formation, from wildtype (WT) and Setdb1 CK-cKO (KO) mice. Gridded rectangles correspond to higher magnification images of cortex in panel **b**. Single and double asterisks correspond to higher magnification images of hippocampus (**a**) and striatum (**b**), respectively. Scale bar = 1 mm (**a** upper panel); 500 μm (**a** lower panel); 100 μm (**b**). **b** (left) cerebral cortex with cortical layers 1–6 as indicated and (right) striatum from section shown in **a**. **c** Bar graphs show quantifications of microglia cell number (left) and immunofluorescent intensity (right) in cortex (Cx, $n = 4$/group, 3/mouse) (left) and hippocampus (Hipp, $n = 6$/group) (right) from WT and KO. Student's unpaired $t$-test, two-sided. $N = 6$ mice/group. Cortex: Intensity: WT, 15.47 ± 1.565; KO, 17.91 ± 1.354; $P = 0.2513$; Average cell counts: WT, 356.0 ± 17.06; KO, 348.5 ± 9.738; $P = 0.7063$. Hipp: Intensity: WT, 21.16 ± 1.035; KO, 26.9 ± 1.272; $P = 0.0057$; Average cell counts: WT, 285.0 ± 16.58, KO, 345.7 ± 11.35; $P = 0.0128$. *$p < 0.05$, **$p < 0.01$. **d** GFAP and Actin loading control immunoblots from 4 brain regions of KO and WT brain, as indicated (Cx cortex, Hipp hippocampus, Str striatum, Cbl cerebellum). **e** (Left) MA-plot show correlation between fold change (KO/WT) and base mean signal of ATAC-seq peaks in purified microglia from Setdb1 wildtype (WT) and knockout (KO) forebrain ($n = 3$/group). Red, $P < 0.05$, Log$_2$FC > 0; blue, $P < 0.05$, Log$_2$FC < 0; grey, $P > 0.05$. (Middle) Genome annotation of up- and down-regulated significant peaks in KO as compared to WT. Blue, promoter; purple, genebody; brown, intergenetic. (Right) Homer de novo motif prediction for differentially regulated peaks. Notice inflammation related transcription factors including Stat2, Arid5a, and NFkB significantly enriched for up-regulated ATAC-seq peaks in KO. Size of the bubble (number in the key) indicates percentage of target sequence with predicted motif (HOMER binomial testing, one-sided, see *Methods*). **f** (Left Bottom): MA-plot shows correlation between gene expression fold change (KO/WT) and abundance. Red, $P < 0.05$, log$_2$FC > 0; blue, $P < 0.05$, Log$_2$FC < 0; grey, $P > 0.05$. (Left top): Clustered heatmap shows differential expression genes ($P < 0.05$) in purified microglia from Setdb1 wildtype (WT) and knockout (KO) forebrain ($n = 3$/group). Mean ± SEM provided as source data. (Middle) ShinyGO analysis show functional enrichment in differential expression genes (KO/WT), FDR < 0.05, Log$_2$FC > 0.5. (Right) Expression of significant interferon-induced genes with normalized FPKM in WT (white bar) and KO (black bar). (Right) Quantification of interferon-induced genes and representative browser shots as indicated. Source data are provided as a Source Data file.

specific *SPRET/EiJ* vs. *C57Bl6/J* Hi-C maps, including mice with neuronal *Kmt1e/Setdb1* ablation, strongly points to epigenomic regulation of (germline-fixed) ERV2 transposable element sequences as a major driver for the B$_2$$^{NeuN+}$-to-B$_2$$^{NeuN+}$ and B$_2$$^{NeuN+}$-to-A$_2$$^{NeuN+}$ contact patterning.

Importantly, despite widespread rewiring and disruption of compartment-specific chromosomal patterning, *Setdb1*-deficient neurons only show few changes in the topologically-associating domain (TAD) landscape, with the notable exception of the *clustered Protocadherin* locus and a few additional genomic sites[27]. Albeit speculative at this point, this apparent phenotypic dichotomy in the Setdb1 deficient mice, with widespread compartment alterations but much more limited changes in TAD landscapes, could reflect non-overlapping regulatory mechanisms governing these two types of chromosomal conformations. For example, phase separation, as a molecular force, shapes compartments, while actively driven loop extrusions present as TADs in ensemble Hi–C data[33]. Thus, the strong interconnectivity of B$_2$$^{NeuN+}$ megadomains, and their regulatory effects on the surrounding A$_2$$^{NeuN+}$ subcompartment, could depend on 'bridges' of heterochromatin-associated protein 1 (HP1) bound to H3K9me2/3-tagged nucleosomes[34] and additional phase separation-promoting mechanisms[35]. To this end, it is notable that, according to the present study, genes showing upregulated expression after neuronal *Setdb1* ablation are primarily affected by loss of A$_2$$^{NeuN+}$–B$_2$$^{NeuN+}$ chromosomal contacts but do not show evidence for direct transcriptional regulation by Setdb1 at the site of the gene. These findings, taken together, suggest that in mature cortical neurons, the mechanisms of transcriptional inhibition include balanced interactions between open (A$_2$$^{NeuN+}$) and repressive (B$_2$$^{NeuN+}$) megadomains. Interestingly, long-range megadomain interactions of H3K9me3-tagged chromatin have recently been implicated in human neurodevelopmental disease associated with instability of short tandem repeats[36].

Our study strongly suggests that the reorganization of higher order chromatin in adult cortical neurons tracks the dramatic expansion of IAPs and other ERV2 transposable elements in *Mus musculus*-derived inbred lines, as compared to the wild-derived SPRET/EiJ inbred strain harboring the 1.5 million year evolutionarily divergent *Mus spretus* genome[25]. Previous studies linked a small subset of these newly inserted ERV sequences to strain-specific differences in gene expression, pathogenic mutations and neomorphisms[37,38], thereby providing 'seeds' for ongoing

phylogenesis in line with the repurposing of even more ancient *Gag* and *Env* sequences into the *Arc* early response and synaptic plasticity gene[39] and the placental *Syncytins*[40]. However, our findings point to much broader role of ERVs for cell-type specific adaptations in genome organization and function, including a partial remodeling of the chromosomal interaction map in mature cortical neurons. The functional importance of proper heterochromatic organization in neurons is underscored by the massive proliferation of IAP particles following the destruction of B$_2$$^{NeuN+}$ in susceptible *Kmt1e/Setdb1* mutant neurons. This, in turn, is likely to cause proteotoxic stress, with the two top ranking GO categories in the differential RNA-seq analysis (Fig. S16), 'Endoplasmic Reticulum (ER) stress response' and 'cytosolic ribosomal protein', reflecting the cell's transcriptional response to excessive translational demand[41]. Furthermore, double-stranded viral IAP RNAs[42] could trigger neuroinflammation, including astrocytosis and microglial activation of immune and interferon response genes (Fig. 5), followed by a more generalized hypermetabolic response in the inflamed brain[43,44].

Importantly, we note that, according to the cell-type specific single DNA molecule sequencing of the present study, *Setdb1*-deficient neurons do not show a detectable increase in IAP/ERV2 de novo insertions (Supplemental Data S11), suggesting the dramatic observed 'neuronal hijacking' by the IAP proviral machinery is primarily driven by un-silencing of preexisting retrotransposon copies as opposed to excessive somatic retrotransposition events. The findings are also of interest from the viewpoint of the potential neurotoxic effects of ERV2-like transcripts and proteins in human and invertebrate neurons, including potential links to tau- and TDP-43 associated neurodegenerative disease[45–49], and recent reports on increased HERV activation in the adolescent non-human primate brain exposed to maternal immune activation during prenatal development[50]. Interestingly, a subset HERVs and other repeat elements are reportedly overexpressed in Alzheimer's brain[46,51]. Furthermore, Alzheimer-related neurodegenerative phenotypes are encountered in mouse models exposed to specific types of HERV-K RNAs[48].

Moreover, HERV-Ks—which, like the murine ERV2s, are members of the betaretrovirus-like supergroup of LTR retrotransposons—have been linked to amyotrophic lateral sclerosis (ALS) or motor neuron disease[51–53] (but see also Garson et al.[54]), including some cases infected with the Human Immunodeficiency Virus (HIV), an exogenous LTR retrovirus[55]. In addition,

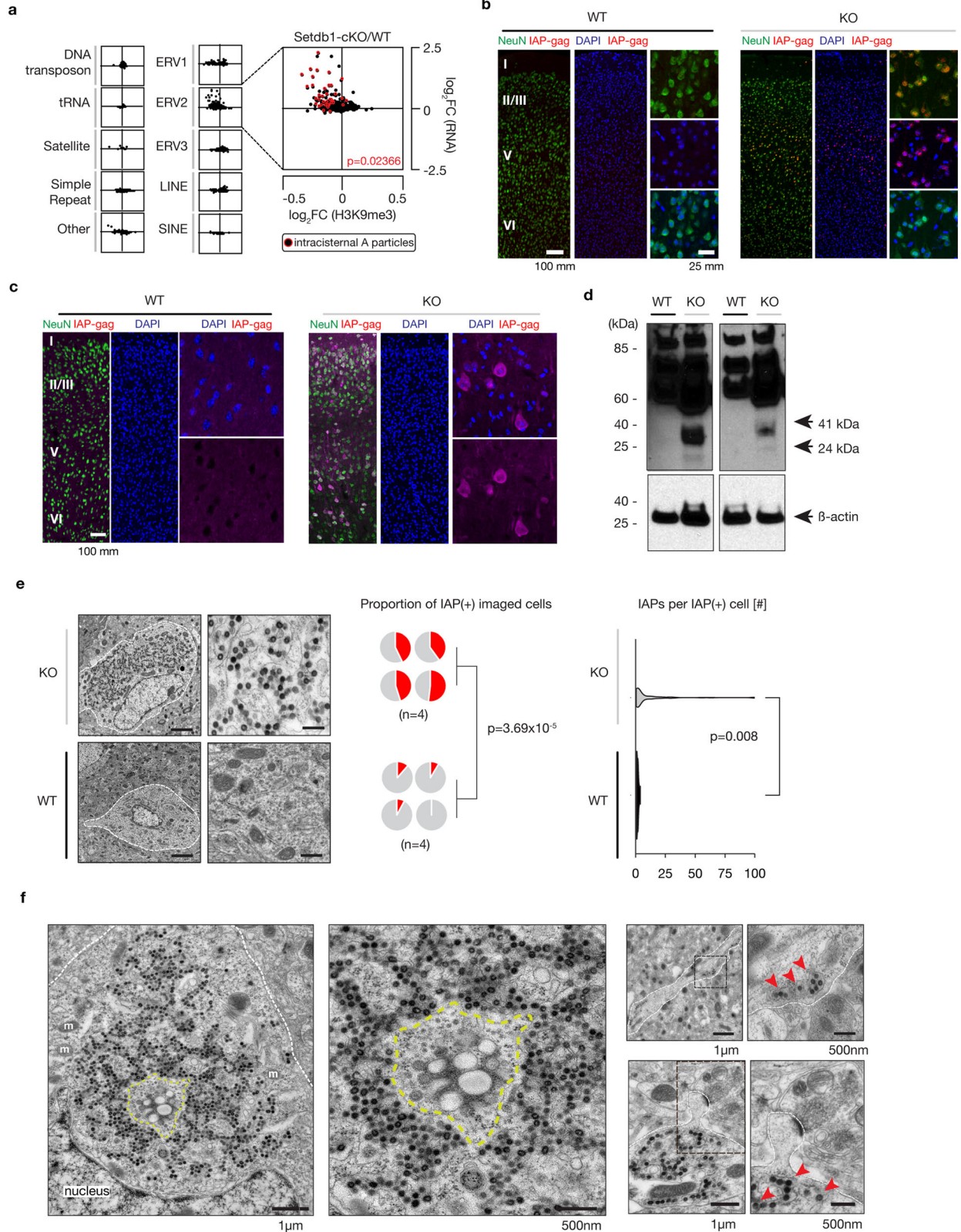

the genetic risk architecture of ALS shows a modest association with genomic loci harboring HERV insertions[53]. Interestingly, we observed that genomic loci harboring HERV-Ks are significantly enriched in *trans* chromosomal Hi-C contacts in human neurons ($p = 1.42 \times 10^{-16}$, Fisher's $2 \times 2$ exact testing), an effect driven primarily by sequences with strong synteny with murine B$_2$ megadomains (Figs. S20 and S21). These findings, taken together,

strongly suggest that compartment-specific enrichment of ERV2/HERV-K sequences occurs in parallel across different mammalian lineages, pointing to a type of heterochromatic organization highly adaptive to species- and strain-specific reconfiguration of the ERV retrotransposon landscape, thereby maintaining a defensive shield to protect neurons from the detrimental effects of LTR retrotransposons.

**Fig. 6 IAP proliferation in Setdb1-deficient neurons. a** Scatterplots of log2FoldChange (H3K9me3 KO vs. WT, x-axis) vs. log2FoldChange (RNA KO vs. WT, y-axis) for each of ten repetitive element categories (Repbase) ($n = 6$/group). The intracisternal A particles (IAPs) are labeled in red and overlaid on the ERV2 plot. The significance of the linear regression is included ($p = 0.02366$). **b** IAP gag RNA FISH signal in coronal sections (30 µm) across multiple cortical layers in KO vs. WT. Counterstained with NeuN (green) and DAPI (blue). **c** Representative immunofluorescence of WT (left) and KO (right) cortical cross sections stained for IAP-gag (pink) and NeuN (green), and DAPI counterstain. Scale bar, 50 µm. Cortical layers, Roman numerals I–VI. **d** Western blotting using the anti-IAP-gag antibody for full-length Gag (73 kDa) and cleaved, processed mature forms (41 kDa and 24 kDa) in WT and KO mouse cortical tissue. b-actin included as loading control. **e** (Left) Cortical layers II/III electron microscopy (EM) images of KO (top) and WT (bottom) neurons imaged at 15 and 30k resolution. (Middle) Quantification of blinded counts of randomly imaged neurons ($n = 50$ or 35) with >1 IAP particle present (IAP + neurons) ($n = 4$ mice/genotype. (genotype [0 IAPs,1+ IAPs]: WT1 [31,4]; WT2 [32,3]; WT3 [47,4]; WT4 [51,0]; KO1 [20,15]; KO2 [21,14]; KO3 [28, 23]; KO4 [26,28]), Student's two-sided t-test ($p = 3.69 \times 10^{-5}$). (Right) Number of IAPs imaged within the IAP + neurons, Wilcoxon sum-rank testing ($p = 0.008$). **f** (Left) Sample neuron at low- and high-resolution with intracisternal proviral packaging and assembly hub marked with yellow dashed line; m mitochondria, nucleus cell nucleus. (Right, top) neuronal dendrites at 15k, 30k resolution. (Right, bottom) Low- and high-resolution images of IAPs at synaptic spine. Red arrows mark individual proviral particles. Scale bars as indicated. Source data are provided as a Source Data file.

## Methods

**Mouse handling**. All mouse work detailed below was approved by the Institutional Animal Care and Use Committee of the Icahn School of Medicine at Mount Sinai. Mice were held under specific pathogen-free conditions with food and water being supplied ad libitum in an animal facility with a reversed 12 h light-dark cycle (lights off at 7:00 a.m.) under constant conditions ($21 \pm 1C$, 60% humidity). All mice were group-housed (2–5 mice per cage). For use in our experiments, mice were reared into adulthood ($\geq 3$ months) and were sacrificed by decapitation following iso-flurane anesthetization according to IACUC guidelines. Following brain removal, cortical dissections were performed on ice aided by a dissection light microscope.

**Cell cultures**. Male germ cells were isolated by the STA-PUT method[56]. A linear gradient was generated using 350 ml of 2% BSA and 350 ml of 4% BSA solutions in the corresponding chambers. Around $1 \times 10^8$ male germ cells were resuspended in 20 ml of 0.5% BSA solution and loaded to the sedimentation chamber. After 3 h of sedimentation in the sedimentation chamber, 60 fractions were collected in 15 ml centrifuge tubes and numbered sequentially 1–60. Cells from each fraction were collected by centrifugation at $500 \times g$ for 5 min and resuspended in 0.2 ml cold medium. An aliquot of each fraction was stained with Hoechst dye (Invitrogen, H3570) and examined thoroughly by eye under phase-contrast and fluorescence microscopes to assess cellular integrity and identify cell types. Fractions containing >80% cells of appropriate size and morphology were pooled as round spermatids.

### Cell-type specific Hi–C

*Fluorescence activated nuclear sorting*. Cortical tissue was prepped for fluorescence activated nuclear sorting (FANS) as follows. Briefly, the dissected tissue was homogenized in a hypotonic lysis solution and fixed in 1% formaldehyde for 10 min at room temperature. The cross-linking reaction was quenched with 125 mM glycine. The nuclei were then purified by centrifugation at $4000 \times g$ and resuspended in a 1:1 solution of the hypotonic lysis solution and a 1.8 M sucrose solution prior to re-centrifugation at $4000 \times g$ to isolate out cortical nuclei. The pellet was then resuspended in Dulbecco's phosphate buffered saline (DPBS) containing 0.1% BSA and 1:1000 anti-NeuN antibody (clone A60, Alexa Fluor 488 conjugated; EMD Millipore Corp., MAB377X). Samples were incubated for 45 min while rotation and protected from light at 4C. DAPI (Invitrogen) was added immediately before sorting to label all nuclei. Sorting was performed at the Flow Cytometry CoRE at the Icahn School of Medicine at Mount Sinai. Nuclei were collected as NeuN+ and NeuN− populations following serial gating and pelleted for downstream experimental processing.

*in situ Hi–C*. Nuclei were digested with 100U MboI, and the restriction fragment ends were labeled using biotinylated nucleotides and re-ligated. After reversal of the cross-linking, ligated DNA was purified and sheared to a length of ~400 bp by sonication, and the biotin-tagged ligation junctions were subsequently pulled down with streptavidin beads (Invitrogen, Dynabeads MyOne Streptavidin T1, Catalog No. 65602) and pre-pared into libraries for next-generation Illumina sequencing (HiSeq 2500). For the SPRET/Eij and matched C57BL/6J studies, the Arima Hi-C kit (Arima Genomics, San Diego) was used according to the manufacturer's instructions.

### Hi–C bioinformatics

*HiC-Pro*. The 'pre-truncation method' using HiC-Pro(v2.9) was used for quality control purposes. Libraries were mapped to the *Mus musculus* reference genome (GRCm38.p5_M13) using bowtie2.2. Artifacts and other common statistics are available as Supplementary Information. HiC-Pro results were piped to Juicer to produce.hic format files using the parser tool 'hicpro2juicebox'. These files, which contain compressed contact matrices at varying resolutions, were then processed into final matrix-balanced normalized contact maps with Juicebox.

*HOMER*. Each read of the paired-end libraries was aligned independently using bwa-mem(v0.7.15), permitting split-mapping to the GRCm38.p5_M13 reference annotation. After mapping, forward and reverse reads were directly supplied to Homer(v4.8) for processing by first, merging the paired-end reads and later, fil-tering out self-ligation (spikes and continuous) artifacts. Files were normalized based on sequencing depth and distance between loci, creating a background model necessary for calculating significant pairwise interactions. Trans- interactions (1 Mb resolution) were similarly calculated, but for significant interactions for loci >200 Mbp apart (-*minDist* 200000000) to disregard intra-chromosomal interactions.

*Subcompartment calling*. We adapted a previously published k-means clustering algorithm[11] to identify subcompartments within our datasets of interests. In short, a 250 kb autosomal resolution map was constructed from.validPairs.hic files gen-erated with HiC-Pro(v2.9). The matrices were normalized with matrix-balancing and processed as observed/expected matrices. Specifically, a subset of inter-chromosomal contact data was extracted using.hic dump and stitched together, with 250 kb loci on odd-numbered chromosomes serially appearing as rows and 250 kb loci on even-numbered chromosomes serially appearing as columns. Odd chromosomal loci were first clustered using the *kmeans* function in R 3.6.0 and RStudio (v1.1.463)., and even chromosomal loci were similarly clustered after transposing the stitched matrix. Several values for the clustering parameter, "k", were tested, ranging from $k = 2$ to $k = 10$ to forcibly organize the genomic loci across these chromosomes into $k$ number of clusters. In NeuN+ and NeuN− (present study), for odd chromosomes, a cluster number of $k = 6$ was determined as the best fit by visual inspection with the generated Hi-C raw read contact matrices; similarly, for even chromosomes, a cluster number of $k = 5$ was deter-mined as optimal. Within each of these cluster sets, clusters consisting of <5% of the genome were disregarded. This ultimately resulted four clusters for the odd and even chromosome sets, that were merged in order of size to represent four sub-compartments for the entire genome. Clusters for the published Hi–C datasets[12–14] were determined based on optimal overlap with the NeuN+ reference, with clusters consisting of <5% of the genome disregarded (Data S4–S7). Subcompart-ment designations for the independent datasets ('A1', 'A2', 'B1', 'B2') were assigned based on the value of the Chi-squared residual when compared to the reference NeuN+ subcompartments.

*Subcompartment characterization*. To appropriately classify these subcompart-ments as active ('A') or inactive ('B'), we leveraged publicly accessible data from the mouse ENCODE project (led by the Mouse ENCODE Consortium). The database is a curation of numerous epigenetic profiles, including transcription factor and polymerase occupancy (http://chromosome.sdsc.edu/mouse/download.html), DnaseI hypersensitivity, histone modifications, and RNA transcription. Strikingly, our results showed a clear dichotomy with respect to overlap with lamina-associated domains[57] wherein two subcompartments exhibited a majority overlap with LADs ('B'), and two did not ('A'). Euchromatin-associated epigenetic features, both active and facilitative (enhancers, CTCF, H3K4me3 (*ENCFF572BJO*), H3K4me2 (*ENCFF647TKR*), H3K9ac (*ENCFF540IHT*), H3K4me1 (*ENCFF224LTF*), H3K27ac (*ENCFF676TSV*), RNAPolII, and H3K36me3 (*ENCFF999HGN*)), showed a near complete overlap with the 'A' subcompartments, while the smaller of the 'B' subcompartments showed moderate overlap with the repressive modification H3K9me3 (*ENCFF101VZW*). Consequently, the derived subcompartments were delineated as 'A' (euchromatic) or 'B' (heterochromatic), and numbered in order of decreasing size, resulting in 'A1', 'A2', 'B1', and 'B2'. Notably, blacklisted ENCODE coordinates (portions of the genome that con-sistently produce anomalous high signal/read counts) accounted for <0.01% of the genomic space encompassed within any of the determined subcompartments; this minimizes the possibility that the signal enrichments determined within these subcompartments are products of aberrant read mapping.

*Hi–C trans and cis calculations*. The aforementioned 250 kb autosomal, matrix-balanced resolution maps of observed/expected Hi–C interaction frequencies were used for downstream analyses; each locus along the axes of the matrix was then labeled according to its NeuN+ subcompartment identifier (A1, A2, B1, B2). For *trans*, Hi–C values from combinations of pairwise interactions involving loci of specific subcompartment designations on different chromosomes were extracted and populated into a vector; for *cis*, these values were extracted for loci on the same chromosome, excluding the diagonal.

*SPRET/EiJ vs. C57BL/6J Hi–C comparison*. Hi-C libraries were generated for sorted NeuN+ and NeuN− populations from SPRET/EiJ and C57BL/6J mice (age- and sex- matched) using the Hi–C Next Generation Sequencing Kit (Arima Genomics) according to the manufacturer's protocol. Next, 250 kb autosomal resolution maps were constructed for using valid pairs generated from the HiC-Pro (v2.9) pipeline following alignment to the *mm10* reference genome. The matrices were normalized with matrix-balancing and processed as observed/expected matrices. Each locus along the axes of the matrix was then labeled according to its NeuN+ subcompartment identifier (A1, A2, B1, B2) and ERV2 count (as determined from UCSC Repeatmasker). A value of 101 was calculated to reflect a locus harboring the 99th percentile of ERV2s as compared to the rest of the genome, and loci with >99th percentile of ERV2s were retained for downstream analyses, generating a reduced Hi-C matrix of $101 \times 101$ loci for each sample. Samples were combined by cell type and strain, resulting in four final matrices representing SPRET/EiJ NeuN+, C57BL/6J NeuN+, SPRET/EiJ NeuN−, and C57BL/6J NeuN−. Statistical significance was performed using Wilcoxon sum-rank testing (paired, two-sided).

*Differential trans- and cis- analysis*. Raw in situ Hi–C matrices (20 kb resolution) were binned into 1 Mb segments genome-wide, and DESeq2 was used to determine windows of significant differential read scores in *trans* ($p_{adj} < 0.05$). Interaction anchors and targets were classified by subcompartment based on percentage of the bin that encompassed subcompartment loci; in the event of a tie, the assignment was made hierarchically in order of subcompartment size, smallest to largest. Expected values of interactions for subcompartment anchor-target combinations were determined from the relative sizes of the subcompartments of interests. Pearson's residuals for differential were calculated by subtracting the residuals of decreased Hi–C interactions.

## Chromatin immunoprecipitation studies

*Native ChIP-seq (NChIP-seq)*. $10^6$ NeuN+ and NeuN− nuclei were pelleted after FANS, resuspended in 300 μl of micrococcal nuclease digestion buffer (10 mM Tris pH 7.5, 4 mM MgCl, and 1 mM Ca$^{2+}$), and digested with 2 μL of MNase (0.2 U/μl) for 5 min at 28 °C to obtain mononucleosomes. The reaction was quenched with 50 mM EDTA pH 8. Chromatin was released from nuclei with the addition of hypotonic buffer (0.2 mM EDTA pH 8, containing PMSF, DTT, and benzamidine). Chromatin was incubated with the appropriate antibodies (anti-H3K9me3 (Abcam, ab8898; 2 μg/$3 \times 10^6$ nuclei), anti-H3K27ac (Active Motif, #39133; 5 μg/$3 \times 10^6$ nuclei), or anti-H3K79me2 (Abcam, ab3594; 5 μg/$3 \times 10^6$ nuclei)) overnight at 4 °C. The DNA-protein-antibody complexes were captured by Protein A/G Magnetic Beads (Thermo Scientific 88803) by incubation at 4 °C for 2 h. Magnetic beads were sequentially washed with low-salt buffer, high-salt buffer, and TE buffer. DNA was eluted from the beads, treated with RNase A, and digested with Proteinase K prior to phenol-chloroform extraction and ethanol precipitation. For library preparation, ChIP DNA was end-repaired (End-it DNA Repair kit; Epicentre) and A-tailed (Klenow Exo-minus; Epicentre). Adapters (Illumina) were ligated to the ChIP DNA (Fast-Link kit; Epicentre) and PCR amplified using the Illumina TruSeq ChIP Library Prep Kit. Libraries with the expected size (~275 bp) were submitted to the New York Genomics Center for next-generation Illumina sequencing (HiSeq 2500, 75 bp, paired-end).

*Bioinformatics*. To perform these bioinformatics analyses, we first aligned the raw.fastq reads to the *Mus musculus* reference genome (mm10) using bowtie2. Only concordant reads were compressed into aligned.bam files then sorted and indexed using the *samtools/0.1.19* suite, and following PCR duplicate removal, were converted into.bed files for further processing.

*Cell-type specific histone profiling*. We performed diffReps[58] differential analysis at a 1 kb resolution for each histone mark of interest at a statistical significance threshold of $p < 0.001$, with NeuN+ histone modification profiles as our treatment group, and NeuN− histone modification profiles as our control group. Significantly called regions (spanning a length of one kilobase of genome or greater) were labeled as NeuN+ enriched or NeuN− enriched according to their log2FoldChange values.

*Genome association testing*. The *Mus musculus* mouse (mm10) reference genome was divided into 250 kb bins using tileGenome() in the GenomicRanges R package. diffReps output files were overlapped with these bins using the sum(countOverlaps) function to retrieve the number of diffReps regions within each 250 kb bin. The two considered variables for the Fisher's testing were (1) the subcompartment identifier of the 250 kb bin and (2) the classification of the 250 kb bin as 'high' or

'low' as it pertained to epigenetic feature enrichment. The threshold distinguishing 'high' and 'low' was set as the 99th percentile of diffReps loci (or repetitive elements) within a 250 kb bin across the autosomal reference genome. Fisher's testing was performed using the *exact2x2* R package. For association with Hi–C interaction changes, bins involved as anchors or targets in the interactions were grouped as the reference genome set. For RNA association testing, no percentile threshold was used for filtering 'high' and 'low'; instead, any bin with altered RNA (either increased or decreased, as indicated) was considered 'high', with bins harboring no altered RNA designated 'low'.

## PacBio SMRT sequencing

*Library preparation*. SMRTbell libraries were constructed from IAPEz-int xGen Lockdown probe-captured DNA for sequencing using a PacBio Sequel II System. In short, genomic DNA was isolated from FACS-sorted NeuN+ or NeuN− nuclei of adult mouse cerebral cortex or round spermatids, using phenol chloroform extraction and was subsequently sheared to ~10 kb using g-Tube microcentrifuge tubes (Covaris, 010145). End-repair, A-tailing, adapter ligation (barcoding) and PCR amplification with the universal primer were performed according to protocol, and fragments were appropriately size-selected using AMPure SPRI Select beads. Samples were then pooled prior to hybridization with biotinylated xGen® Lockdown® designed against the consensus sequences (dfam.org) of IAPEzi or IAPEY3-int (control) (Integrated DNA Technologies). Streptavidin A1 beads were used to capture hybridized fragments, which were subsequently amplified prior to SMRTbell library preparation. Following AMPure purification and size selection, SMRTbell templates were annealed and bound to the barcoded libraries, which were submitted for PacBio Sequel II HiFi sequencing (Genewiz, 30 h movies). Oligonucleotides used for this preparation are included as Supplementary Tables.

*Bioinformatics*. Bioinformatics processing was performed with SMRT Link v5.1.0, run with default parameters, unless otherwise indicated. Files were first demultiplexed with *lima* and circular consensus sequences (CCS) were generated with *ccs* for filtered sequences with matching paired-end adapters. CCS reads were then mapped to an IAPEzi consensus sequence (dfam.org) using *pbalign*. Passing reads were subsequently mapped to the mouse reference genes using bwa split mapping and filtered based on mapping score (=60). Reads were then assigned as autosomal IAPEzi or autosomal non-IAPEzi IAP using the GenomicRanges subsetByOverlaps() function. The other reads, those not mapping to IAPs denoted in the reference genome, were then blasted against the IAPEzi consensus sequence and IAP masked reference genome in parallel using BLAST + ((v2.7.1); reads with higher bit scores for the IAPEzi blast than the reference genome were defined as de novo. Full-length de novo insertions were defined as fragments representing >5833 bp ($0.9 \times 6481$ bp), with average full-length IAPEzi lengths being 6481 bp (dfam.org).

## Cortical transcriptomics

*RNA-seq*. Total RNA was first extracted from the prefrontal mouse cortices ($n = 12$, 6WT/6KO), and prepared with RNeasy Lipid Tissue Mini kit (with on-column DNase1 treatment). The quantity and quality of RNA was checked using a bioanalyzer (Agilent RNA 6000 Nano Kit). In all, 1.5 μg of total RNA from each sample was submitted to the Genomics CoRE Facility at Mount Sinai for RNA-seq library generation with RiboZero treatment, and were subsequently sequenced on the Illumina HiSeq 2500, 100 bp, paired end.

*Bioinformatics*. Read pairs were aligned to the *Mus musculus* mouse reference genome (mm10) Tophat2 short-read aligner. Reads were counted using HTSeq against the Gencode vM4 Mouse annotation. Genes were filtered based on the criteria that all replicates in either condition must have at least five reads per gene. On the resulting filtered transcript, a pairwise differential analysis between Setdb1 conditional mutant vs control cortex was performed using the voomlimma R package[8,9] which converts counts into precision weighted log2 counts per million and determines differentially expressed genes using a linear model. Significantly differentially expressed genes were identified using a cutoff of Benjamini–Hochberg adjusted $p$-value < 0.05. Discordant pairs were extracted from aligned.bam files using *samtools* with the following command: samtools view –b –F 2 and reads identifiable with specific flags were quantified and categorized.

*Subcompartment neighbor testing*. Subcompartment bins (250 kb) were first determined and reduced using GenomicRanges into contiguous segments of similar identity. The percent of all boundaries by different combinations of subcompartment boundaries were calculated as the observed values. For expected values, genomes were reconstructed using the 250 kb blocks and reduced as above. The percent of boundaries by different combinations of subcompartment boundaries were calculated × 100 permutations using regionER resampleRegions (mean + standard deviation).

*GO enrichment*. Gene ontologies for gene sets of interest were determined using the ClueGO (v2.5.1) application through Cytoscape (v3.6.1). All available ontologies/pathways (CORUM, Chromosomal-Location, GO, INTERPRO, KEGG, REACTOME, WikiPathways) were considered, and the chosen network specificity was

'medium', displaying only pathways with pV < 0.05. The statistical testing performed was an enrichment (right-sided hypergeometric test) analysis with Benjamini-Hochberg pV correction. All other parameters were left as default parameters, including GO Term Grouping.

## Microglial studies

*Microglia isolation.* Single-cell suspension from adult brain tissues was prepared using Miltenyi's Adult Brain Dissociation Kit (Miltenyi Biotec, 130-107-677) and Debris Removal Solution Kit (Miltenyi Biotec, 130-109-398) according to manufacture instruction with minor modifications. In brief, total brain tissues, except olfactory bulb and cerebellum, were collected quickly and washed with ice cold 1x HBSS. After chopped into small pieces using sharp blade, the brain tissues were transferred into the C-tube containing 1950 μl of Enzyme mix 1 and 30 μl of Enzyme mix 2 from the kit and incubated on Miltenyi's gentle MACS Octo Dissociator with Heaters using program 37 °C _ABDK_1 for 30 min. Afterwards, the digested tissue homogenate gently went through fire polished glass pippette 10 times, and then passed through a 70 μm cell strainer. Cells were collected via centrifuged at 300×g for 10 min at 4 °C and resuspended in ice cold 1x HBSS. Debris Removal Solution was then added and overlayed with ice cold 1x HBSS. After centrifugation, three phases in the tube were clearly visualized, from top to bottom: 1x HBSS solution, debris and myelin layer, and single-cell suspension. Discard the top two phases completely, wash the cells in 1x HBSS, centrifuge to collect the cells, and then resuspended the cell pellet in 1 ml of 1x HBSS containing 1% fetal bovine serum. In order to minimize the unwanted microglia activation, single cell suspension was incubated in Fc receptor blocking Reagent (Miltenyi Biotec, 130-092-575) for 10 min, and then the cell suspension was incubated with CD11b-microbeads (Miltenyi Biotec, 130-093-634) for 10 min at 4 °C in the dark, followed by positive selection with LS separation column (Miltenyi Biotec, 130-042-401). Flow cytometry analysis was performed to check the cell purity. The three different groups of cells (no enrichment, target and non-target cell population) were incubated in CD11b-FITC Monoclonal Antibody (M1/70) (eBioscience, 11-0112-81) at 1:2000 dilution for 30 min at 4 °C in the dark. Stained cells were examined using the Beckman flow cytometer.

*Immunostaining.* The coronal sections (30 μm) were washed with 1x PBS containing 0.5% Triton X-100 for 10 min, and then blocked with 1x PBS containing 0.5% Triton X-100 and 5% goat serum, followed by primary antibodies incubation for anti-Iba1 (Abcam, ab178846, 1:1000 dilution), anti-GFAP (Abcam, ab4674, 1:4000 dilution), anti-NeuN–Alexa555 (Sigma-Aldrich, MAB377A5, 1:2000 dilution), and Tomato-lectin (Vector, DL-1174, 1:100 dilution) at 4 °C overnight. Sections were incubated with fluorescence-labeled secondary antibodies (goat anti-chicken-Alexa488 (Jackson, 103-545-155, 1:1000 dilution); goat ani-rabbit-Alexa594 (Jackson, 111-585-003, 1:1000 dilution) at room temperature for 2 h in the dark. Images were captured with a fluorescence microscope (Olympus VS120). Analysis was performed using NIH Image J software.

*RNA-seq.* Total RNA was extracted by using Direct-zol RNA MicroPrep (Zymo research, R2060). Library was prepared using QIAseq FastSelect RNA Removal Kit (Qiagen, 333180-24) and QIAseq Stranded Total RNA Lib Kit (Qiagen, 180743) following manufacturer's instruction. In brief, 200–500 ng total RNA was used for each reaction. 1ul rRNA removal reagent was added into 28 μl of RNA sample, plus 5 μl 5×RT Buffer, followed by incubation for 3 min at 95 °C, and then went through stepwise annealing using PCR programing. After fragmentation and rRNA removal, reverse transcription, second-strand synthesis, end-repair, A-addition, and strand-specific ligation with selected adapter (1:25 dilution) were performed, followed by CleanStart library amplification. Library DNA was then purified and size selected for around 500 bp fragments, and checked with Qubit and Agilent 4200 Tapestation.

*ATAC-seq.* Cells were collected and nuclei were extracted by douncing directly in lysis buffer (0.32 M sucrose, 5 mM CaCl₂, 3 mM Mg(Ace)2, 0.1 mM EDTA, 10 mM Tris-HCl pH = 8, 0.1% NP-40). In all, 50,000 nuclei were used for each reaction. Library was prepared using TruePrep DNA Library Prep Kit V2 for Illumina (Vazyme, TD501-02) following manufacturer's instruction with minor modifications. In brief, nuclei were treated with Tn5 at 37 °C for 30 min, and then DNA was purified with MinElute Gel Extraction Kit (Qiagen, 28604), and elute in 25 μl of EB buffer. PCR was then setup by the mixture of 20 μl DNA sample, 10 μl TAB, 5 μl PPM, and selected adaptors (5 μl N5XX and 5 μl N7XX) (TD202), followed by PCR amplification. Library DNA was purified by using SPRIselect beads (Beckman, B23318) for 1:0.55 followed by 1:1.15 two steps size selection, and then checked with Qubit and Agilent 4200 Tapestation.

## Microglial data analysis

*RNA-seq.* Sequencing was performed on Illumina XTen (PE150). Before analyzed the data, FastQC was first used for quality control analysis, and Trim-galore was used to remove low-quality reads. Paired-end clean data was aligned to reference genome (M. musculus, UCSC mm10) using Tophat2 v2.1.1. Samtools v1.9 was used to sort and build the alignment files index. FeatureCounts v1.6.3 from subread package was used to get gene expression level counts by determining the number of

reads mapped to gene exons, and differential analysis was generated by DESeq2. Significant genes (log₂FoldChange > 0.5, $P_{adj} < 0.05$) were extracted for Gene Ontology Enrichment Analysis by using ShinyGO v0.61 (http://bioinformatics.sdstate.edu/go/). Deeptools was used to generate normalized bigwig and visualized on IGV.

*ATAC-seq.* Sequencing, FastQC, and data cleaning were performed as described above. Paired-end clean data was then aligned to reference genome (M. musculus, UCSC mm10) using Bowtie2 v2.3.4.3. Samtools v1.9 was used to sort and build the alignment files index after removing duplications. MACS2 was used for peak calling (--shift -100 --extsize 200 --nomodel). Diffbind was used to generated differential change peaks. R packages "ChIPseeker", "clusterProfiler" "org.Mm.eg.db" and "TxDb.Mmusculus.UCSC.mm10.knownGene" were used for peak annotation. Significant differential peaks that are located on gene promoters were extracted to generate Gene Ontology Enrichment Analysis by ShinyGO v0.61 (http://bioinformatics.sdstate.edu/go/). Deeptools was used to generate normalized bigwig and visualized on IGV.

**RNA and DNA in situ hybridization of IAP-gag probes**. Coronal mouse brain slices (10 μm) were collected using a freezing microtome and mounted onto Superfrost plus slides (Fisher). The slides were processed per the RNAscope Multiplex Fluorescent v2protocol (Advanced Cell Diagnostics). Briefly, the tissue sections were treated with protease, hydrogen peroxide, and incubated with either sense probes for DNA FISH, or anti-sense probes for RNA FISH, targeting 340 nt of the IAP-gag sequence (bp 1547-1914)[59] for 2 hours at 40 °C. Amplifier sequences were polymerized to the probes and treated with Opal 570 dye. The slides were incubated in NeuN−AlexaFluor488 antibody (1:200 in 1xPBS; MAB377X; EMD Millipore) for 2 h at room temperature and cover-slipped with DAPI Fluoromount (Southern Biotech). Imaging was performed on a Zeiss LSM780 confocal microscope.

## Protein expression

*Immunohistochemistry and western blotting.* For GFAP IHC, coronal sections from perfusion-fixed (by phosphate-buffered 4% paraformaldehyde) adult mouse brains coronal sections from perfusion-fixed (by phosphate-buffered 4% paraformaldehyde) adult mouse brains were processed for anti-GFAP immunoactivity and detected with diaminobenzidine (DAB) using the ABC kit (VectorLabs) according to the manufacturer's protocol.

For GFAP immunoblotting, protein was extracted from homogenized adult mouse cortical (Cx), hippocampal (Hipp), striatal (Str), and cerebellar (Cbl) tissue, and 100 μg total protein was loaded in each lane. The membrane was blotted with either anti-GFAP (Abcam, ab4674, 1:8000 dilution) or anti-actin (Sigma, A2066, 1:2000 dilution) and probed with either goat anti-chicken IgY H&L (Abcam, ab6877, 1:10,000 dilution) or HRP-conjugated Affinipure goat anti-rabbit IgG(H + L) (Proteintech, SA00001-2, 1:10,000 dilution) for 1 h at room temperature prior to detection.

For IAP immunohistochemistry, adult conditional Setdb1 mutant and control mice - approximately 6 months old - were anesthetized with a terminal intraperitoneal injection of a ketamine/xylazine mixture (IP: 200 and 30 mg/kg, respectively). Transcardial perfusion was performed with 100 ml of 10% sucrose followed by 200 ml of 4% paraformaldehyde in PBS. Brains were removed and placed in 4% formaldehyde overnight at 4 °C, followed by incubation in 30% sucrose until isotonic. After embedding in OCT compound (Tissue-Tek), the brains were cut on a freezing microtome (Leica SM2010 R) into 30 μm coronal sections and placed in 1x PBS. Staining for IAP protein was performed as follows: coronal sections containing prefrontal cortex were blocked and permeabilized with 10% BSA and 0.05% Triton X-100 in 1x PBS for 1 h at room temperature, followed by incubation with the rabbit anti-IAP antibody (see paragraph above) (1:100 in 1x PBS; and 0.01% Triton X-100 overnight at room temperature. The sections were washed for 5 minutes in PBS followed by incubation in anti-rabbit Alexa Fluor 647 secondary antibody for 1 h at room temperature. Sections were washed briefly in PBS before being mounted on Superfrost Plus slides (Fisherbrand) with DAPI Fluoromount-G media (SouthernBiotech). Imaging was done using a Zeiss CLSM780 upright microscope.

For IAP immunoblotting, protein was extracted from homogenized adult mouse cortical tissue and 100 μg total protein was loaded in each lane. The membrane was blotted with rabbit anti-IAP antibody (a gift from Dr. Bryan R. Cullen, Duke University, 1:10,000 dilution) and probed with goat anti-rabbit HRP (Invitrogen, #31460; 1:5000 dilution) for 1 h at room temperature prior to detection.

**Electron microscopy**. Adult mice (N = 4 conditional Setdb1 mutants and N = 4 control animals) were anesthetized and perfused using a peristaltic pump at a flow rate of 35 mls/min with 1% paraformaldehyde/phosphate buffered saline (PBS), pH 7.2, and immediately followed with 2% paraformaldehyde and 2% glutaraldehyde/PBS, pH 7.2 at the same flow rate for an additional 10–12 min. The animal's brain was removed, and placed in immersion fixation (same as above) to be post-fixed for a minimum of 1 week at 4 °C. Fixed brains were sectioned using a Leica VT1000S vibratome (Leica Biosystems Inc., Buffalo Grove, IL) and coronal slices (400 μms)

containing the frontal cortex were removed and embedded in EPON resin (Electron microscopy Sciences [EMS], Hatfield, PA). Briefly, sections were rinsed in buffer, fixed with 1% osmium tetroxide followed with 2% uranyl acetate, dehydrated through ascending ethanol series and infiltrated with EPON resin (EMS). Sections were transferred to beem capsules, and heat polymerized at 60 °C for 48–72 h. Semithin sections (0.5 and 1 μm) were obtained using a Leica UC7 ultramicrotome, counterstained with 1% Toluidine Blue, cover slipped, and viewed under a light microscope to identify and secure the layers of interest (L11 – L1V). Ultra-thin sections (80nms) were collected on copper 300 mesh grids (EMS) using a Coat-Quick adhesive pen (EMS), and serial sections were collected on carbon-coated slot grids (EMS). Sections were counter-stained with 1% uranyl acetate followed with lead citrate and imaged on a Hitachi 7000 electron microscope (Hitachi High-Technologies, Tokyo, Japan) using an advantage CCD camera (Advanced Microscopy Techniques, Danvers, MA). Images were adjusted for brightness and contrast using Adobe Photoshop 11.0.

**Statistics**. Statistical testing manually performed for the studies included in this manuscript are compiled here; statistics incorporated within bioinformatic processing algorithms are discussed in the relevant, aforementioned sections. For Fig. 1e, to analyze subcompartment associations of histone modification enrichments in NeuN+ vs. NeuN−, we used Fisher's $2 \times 2$ testing. For Fig. 1g, to perform correlations among Hi-C *trans* interactions across replicates within and across cell types, we used Pearson's correlation testing. For Fig. 2a, to analyze subcompartment association of genome repeats, we used Fisher's $2 \times 2$ testing. For Fig. 2c, to compare observed/expected Hi–C frequencies among ERV2-rich loci in SPRET/EiJ vs. C57BL/6J, we used Wilcoxon sum-rank testing. For Fig. 3b, to analyze qPCR amplification of IAPEzi-biotinylated oligo capture products using select primers, we used Student's *t* testing, paired. For Fig. 4b, to analyze differential Hi-C interactions in KO vs. WT neurons by subcompartment designations, we used Pearson's residuals. For Fig. 4c, to analyze the relationship of differential Hi–C interactions in KO vs. WT with genome repeat hotspots, we used Fisher's $2 \times 2$ testing. For Fig. 4d, to analyze subcompartment associations of upregulated genome repeats in KO vs. WT, we used Fisher's $2 \times 2$ testing. For Fig. 4e, to analyze the spatial relationship of subcompartment megadomains genome-wide, we used permutation analysis testing. For Fig. 4f, to analyze subcompartment associations of DEGs in KO vs. WT, we used Fisher's $2 \times 2$ testing. For Fig. 5c, to compare Iba-1(+) cell counts and intensities in mouse cortex and hippocampus, we used Student's *t* testing, unpaired. For Fig. 6a, to study the association of H3K9me3 in KO vs. WT with RNA changes, we used linear regression analysis. For Fig. 6e, to study both the proportions of IAP(+) cells in WT vs. KO neurons and the proportions of IAPs per IAP(+) cell in WT vs. KO, we used Student's *t*-testing, unpaired. For Fig. S5B, to analyze NeuN+ vs. NeuN− *trans* interactions per subcompartment (O/E distribution), we used Student's *t*-testing, paired. For Fig. S19, to compare human subcompartments from Rao et. al. with syntenic murine NeuN+ subcompartment coordinates, we used Pearson's residuals. For Fig. S20, to study the associations of genome repeats with syntenic human subcompartments, we used Fisher's $2 \times 2$ testing.

**Reporting summary**. Further information on research design is available in the Nature Research Reporting Summary linked to this article.

## Data availability

The data that support this study are available from the corresponding authors upon reasonable request. The sequencing data for genome-scale analysis (Hi–C, ChIP-Seq, RNA-Seq, and PacBio SMRT-Seq) generated in this study have been deposited in NCBI's Gene Expression Omnibus (GSE168524). Other publicly available datasets used in this paper are available at Mouse Encode Project (http://chromosome.sdsc.edu/mouse/download.html, SRP154319, GSE125068, GSE96107, and GSE99363. Source data are provided with this paper.

## Code availability

Scripts used for bioinformatics analyses are available at https://github.com/sandchand5/hic-neuron-erv (https://doi.org/10.5281/zenodo.5549985).

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

## Acknowledgements

The authors thank all members of the Akbarian lab for constructive comments and discussions; E.X., C.R. and A.C. for generous access to sequencing equipment; L.W. and staff at the New York Genome Center for logistical support; M.S. and D.S. for guidance on the PacBio SMRT-Seq protocol; and L.M. for his insights into human retrovirology. Nuclei sorting was performed at the Flow Cytometry CoRE at the Icahn School of Medicine at Mount Sinai. Microscopy preparation and imaging was performed at the Microscopy CoRE at the Icahn School of Medicine at Mount Sinai. RNA-Seq library preparation and next-generation sequencing was performed at the Genomics CoRE at the Icahn School of Medicine at Mount Sinai. We thank Dr. B.R. Cullen (Duke University) for the anti-IAP antibody gift. This work was supported by grants R01MH106056 (S.A.), R01MH117790 (S.A.), and P50MH096890 (S.A.), National Natural Science Foundation of China 81971272 (Y.J.), and T32-AG049688 (S.C.).

## Author contributions

S.C., B.J., B.K., Y.D., Y.Z., Y.Z., L.K.B., P.R., C.J.P., D.J., E.A.-P. and M.I. performed experiments; S.C. and S.A. conceived and designed experiments; S.C. performed statistical analyses; S.C., S.E.-G., Y.-H.E.L., Y.D., B.J.L. and H.L. provided resources. M.E. and L.S. performed bioinformatics and genomic analyses; Y.J. and S.A. supervised the research; S.C. and S.A. wrote the paper, with contributions from all co-authors.

## Competing interests

The authors declare no competing interests.
