## [Peer Review File · Nature Communications]

REVIEWER COMMENTS

Reviewer #1 (Remarks to the Author):

The elegant study by Chandrasekaran and colleagues claims that endogenous retroviruses (ERVs), in particular ERV2, are essential partners in maintaining the three-dimensional organization of large segments of chromatin (megabase scale) of neurons in the mouse cerebral cortex. The authors show that targeted deletion of the retrotransposon silencer *Kmt1e/Setdb1* within neurons, using the Cre-LoxP approach, leads to disintegration and rewiring of the chromosomal interactions. This is accompanied by infiltration of ERVs (particularly IAP) into dendrites and dendritic spines in the neurons, as well as robust activation of interferon and cytokine signaling pathways in cortical microglia.

This manuscript was a challenging read to this reviewer but the originality of the approach, as well as the novelty and strength of the findings, and the author's interpretations, made it a worthwhile exercise. They present significant results that would likely influence the field of genomics, and most definitely will have an impact on the emerging field of ERVs. Indeed, not that long ago, ERVs were dismissed as junk DNA. This manuscript adds an outstanding set of observations that points to the biological relevance of ERVs. It could also be of high interest to the field of neuroinflammation.

The results that are directly focused on ERVs and chromatin analysis are well-described and superbly presented (figs. 1, 2, 3, and several supplemental figures and tables). The authors use state-of-the-art techniques to study 3D genome organization. The methods (in the supplement) are clearly written.

My overall enthusiasm for this manuscript is somewhat dampened by a few weak aspects listed below. I think addressing them will result in an even more impactful study.

Suggested improvements:

-- Introduction: it consists of a single paragraph that mostly summarizes the results of the manuscript itself. I strongly recommend that the authors add more substance to this section of the manuscript, with the goal of making their study more accessible to a larger audience. Specifically, it would be quite helpful to add some sentences about the genomic technology utilized in the study, which can go immediately after the sentence starting with "However, the relationship..." (pg. 3, line 8). At minimum, Hi-C needs to be described as the state-of-the-art technology to evaluate chromosome conformation capture (3C). The authors need to cite the seminal paper by Lieberman-Aiden et al (2009, *Science* 326:289-293).

--also, Introduction: a few sentences describing previous studies on ERVs in the context of neuroscience, and particularly neuroinflammation, would be a useful addition. Here, I suggest they move a few of their supplementary references to the main text, such as Sharif et al. (2013, *Philos Trans R Soc Lond B Biol Sci* 368:20110340) and Sankowski et al. (2019, *PNAS* 116:25982-25990). They may also want to add the reviews by Romer (2021, *Front Neurosci* 15:648629) and Hantak et al. (2021, *Trends Neurosci*, DOI:<https://doi.org/10.1016/j.tins.2020.12.003>).

-- Statistical tests: the methods section lacks a separate subsection about statistics. This omission needs to be corrected. Moreover, there are numerous p values within the Results section but very few indicate which test was used. An example of good practice is on pg.4, line 20: "p=3.36x10E-41, Fisher's 2x2 exact test". The authors need to use this format across the whole manuscript.

-- The figures are clearly laid out, except for a few aspects:

Fig.1d: I suggest that the panels are placed horizontally so that the NeuN+ data is at the left and the NeuN- data is at the right.

Fig.1e: I recommend they place the schematic at the left, the NeuN+ data at the middle, and the NeuN- data at the right; making sure that the A1, A2, B1, and B2 labels are written explicitly for each dataset. This layout avoids the somewhat incorrect repetition of the schematic in 1e.

Fig.1g: I suggest the color scale for Pearson's correlation is modified, so 0 is represented in blue and 1 by red. This color scale is coherent with the rest of the manuscript. Parenthetically, the color scheme going from dark blue to light blue led this reviewer to think that all the values were near zero. One needs careful examination to realize that the values are actually representing a larger range.

Fig.1g: it is unclear what the ($p < 10E-50$) refers to (inside the figure or in the figure legend). What samples were being compared? Which test was used in this case? I recommend the authors do not add the number in the figure itself, but that they add the information of what test was run in the figure legend. Indeed, they need to make sure that the statistical test is stated for every comparison in the manuscript.

Fig.1h: the graphs showing the Hi-C trans interactions need a Y scale, similar to the scale in Fig.1f. I think a single Y scale at the very left would be sufficient.

Fig.2e: it is unclear why each red arrowhead needs letters, as they do not add extra information.

Fig.4a: the GFAP panels are underwhelming. Based on the Western blot information in Fig.4b, one would expect that the GFAP staining is significantly increased in the KO cortex. With a generous eye, one can almost see more blue staining in the deep cortical layers, but I think these panels need to be improved. The authors also need to label the cortical layers (as they do in Fig.5b and 5c).

Fig.4b: it would be great if the authors can show GFAP-stained tissue in addition to the Western blot. I understand they already show data for cortical astrocytes in 4a (with the caveats I just mentioned), but a striatal section (which would apparently show stronger staining in the KO) and a hippocampal section (in which both are similar) would be a nice addition and quite useful to astrocyte aficionados.

Reviewer #2 (Remarks to the Author):

This manuscript analyzed Hi-C data in the mouse cortex. The reported B2-NeuN+ subcompartments, which is enriched with trans contacts in NeuN+ neurons, and also enriched with H3K9me3 and ERV2 repeats. Based on these observations, the authors hypothesized that such ERV2 loci form a subcompartment that is dependent on heterochromatin. The tested Setdb1 KO mice and observed decreased trans contacts, increased viral load, and increased microglia inflammatory signal. The conclusion is that there is a neuronal 3D genome compartment architecture to protect the neurons from retroviral toxic effects. I think the results relevant to retroviral reactivation in Setdb1 KO mice is interesting, although kinda expected. However, I am not convinced about the existence and the function of a ERV trans compartment solely based on Hi-C data, as the authors suggested. Additionally, the authors should substantially simplify their figures, most of which are unnecessarily busy.

1. Simple compartment A/B calling is rather stable reflecting euchromatin/heterochromatin, but further classification at subcompartment level is much more variable between experiments. Most importantly, the subcompartments are only computational separation of the genome only based on Hi-C data. There is not a general consensus what the A1/A2/B1/B2 represent in biology molecularly/chemically besides some non-decisive correlation with various histone marks. Therefore starting this paper by comparing "B2" in NeuN+ cells to the "B2" in NeuN- cells is meaningless and an over-interpretation of the Hi-C data; the "B2" in two cell types are completely different things. This is evident from Fig. 1, there are 104M of B2-NeuN+ but 396M of B2-NeuN-, the B2-NeuN- are actually enriched with some euchromatin features including CTCF and H3K4me1.

2. If we only focus on B2 in NeuN+, how consistent is B2 calling among all biological replicates (2F/2M

mice)? The authors should provide a direct comparison instead of using Fig.1g.

3. The authors claim from Fig 1g about trans contacts that: (i) the 4 mice are similar; (ii) there are differences between NeuN+ and NeuN-. However, the most recognizable "hotspots" of the trans-contacts shared between 4 mice NeuN+ are also hotspots in NeuN- cells, they do not look to be cell type specific.

4. Fig. 1h shows the best recognizable hotspots from Fig. 1g but try to show they are at different locations on NeuN+ and NeuN-. For example, the hotspot on chr4 in NeuN+ and NeuN- seems to be the same in Fig. 1g, but Fig. 1h shows that the chr4 trans-contact hotspots are different between NeuN+ and NeuN- cells? However, if hotspots are truly cell type specific, why do they appear on the same chromosomes? This is a paradox.

5. Relevant to #3 and #4. mm10 is not perfect at the repeats. Calling trans-contact from Hi-C can be wrong if there are translocation/copy number variants/structure variants. Therefore, interpreting trans-interactions at transposons is especially risky. How can the authors exclude the possibility that the observed trans-contacts domains are not due to those reasons? Unannotated transposition may actually explain the paradox mentioned in #4. For example, if there is active viral insertion in NeuN+ cells, that will explain the gain of trans-contacts (due to more copies); some copies may gain H3K9me3 and lead to increased the H3K9me3 ChIP-seq signal. These can be mis-interpreted as "subcompartment".

7. I therefore think the authors cannot claim "subcompartment" solely based on Hi-C. Independent experiments (e.g. FISH) is necessary.

8. I have no problem with Fig. 4-5 but they do not support the subcompartment.

9. The discussion about phase separation is speculation. The authors do not have experiments showing that HP1 or heterochromatin is organizing subcompartment via phase separation.

Reviewer #3 (Remarks to the Author):

In this study Chandrasekaran and colleagues report the existence of four large chromosomal subcompartments (A1 and A2 in eucromatin, and B1 and B2 in heterochromatin) in cells from the adult cerebral cortex through Hi-C analysis. They next demonstrate a higher frequency of trans interactions at the B2 megadomain in NeuN+ cells compared to NeuN- cells. This B2 megadomain shows an enrichment for lamin associated domains (LADs) and H3K9me3 compared to the A1, A2 and B1 megadomains, and is enriched in endogenous retrovirus subtype ERV2 with retrotransposition capacity. Loss of Kmt1e/Setdb1, a histone H3K9 methyltransferase involved in retrotransposon silencing, caused the disintegration of the megadomain and led to proviral assembly of ERV2/Intracisternal-A-Particles (IAPs) that infiltrate into dendrites and spines. The authors hypothesize that the expression of ERV2 retrotransposons elicits a cellular stress response that may be responsible for the severe neuroinflammation observed in Setdb1 deficient mice. Overall, the study presents a novel and intriguing concept with important implications. I have several recommendations to enhance the presentation of the results and clarify the text and figures.

Major points:

1) I may be overinterpreting the main findings, but I believe that the study suggests that neurons have evolved a megadomain architecture that play a defensive role (referred by the authors as "adaptive") against ERV2 intragenomic parasites. When this architecture is disrupted by the loss of the Setdb1 safeguard, the de-repressed domain unleashes ERV proviruses with strong tropism within mature neurons, leading to neuropathology. Although this result is presented in the frame of the

characterization of a mouse strain (Setdb1 KO) that does not model any particular disorder, chromatin de-repression has been proposed as a mechanism of etiopathology in various rare neurodevelopmental disorders, and the erosion of epigenetic profiles and neuroinflammation are features of numerous neurological conditions and the aged brain. The authors could highlight the relevance of their findings by investigating if ERV2 derived transcripts are upregulated in transcriptome datasets for some of these conditions, which could further underscore the defensive role of Setdb1 (and likely other epigenetic enzymes).

2) Recent studies have generated Hi-C datasets from mouse brain with more resolution than the data presented here. The authors should confirm that their conclusions regarding the high incidence of trans interaction in the B2 compartment, etc, are reproduced in these other Hi-C studies. They may also take advantage of the data generated in different developmental stages (from stem cells to mature neurons; see, for example, the studies by Bonev et al. 2017 and Fernandez-Albert et al, 2019) to further explore the cell-type-specific differences in the B2 compartment described here.

3) Given the cellular complexity of the adult cerebral cortex, both the NeuN+ and the NeuN- populations contain a mix of cell types. However, the NeuN- population is expected to be more heterogeneous than the NeuN+ population, and likely contain more divergent cell types. Can this heterogeneity explain the weaker contacts observed in the NeuN- population? Related to the previous point, the authors could take advantage of published Hi-C data for specific cell types to explore the cell type specificity of B2 presented in the first part of the manuscript.

4) The authors could demonstrate that ERV2 transposable elements are differently H3K9me3-enriched in the NeuN+ and the NeuN- populations by ChIP assay. This would further support a direct role of H3K9me3 (Setdb1 substrate) in neuron-specific B2 megadomain interactions. In the subsequent analysis of Setdb1 KO the authors only analyze the changes in NeuN+ population and they do not analyze the consequences of Setdb1 depletion in the NeuN- population.

5) In the comparative analysis of the two mouse strains (Spret/EiJ and C57/BL6J), the authors do not show the subcompartmentalization between A1, A2, B1 and B2 in cis and trans (similar to what it is presented in Fig. 1C) to verify that the B2 NeuN+ population continues to have a specific compartmentalization in Spret/EiJ background although, the interaction frequency is lower compared to the strain C57/BL6J.

6) The transcriptional analysis of the Setdb1 KO mice focused on the changes related to ERV2, but there are also many transcriptional changes in the A2 megadomain. Are DEGs in the A2 megadomain also associated with H3K9me3 enrichment? It would be interesting to show a more detailed dissection of the transcriptional results, exploring the type of DEGs up-regulated in each megadomain.

7) In page 8, the authors conclude that the analysis of DEGs in Setdb1-KOs "potentially indicates a hypermetabolic state in response to an inflammatory stimulus". Is this a cause or a consequence of the epigenetic alterations reported in these animals? The authors could propose a causal relationship between their various observations in these animals in the Discussion section.

8) In general, the authors should revise the figure legends. Some panels are very complex and their succinct descriptions do not facilitate their comprehension.

9) While some heatmaps benefit from using a divergent color scale, such as the one presented in Fig. 3b in which the neutral color is at the middle of the scale. However, other heatmaps work better using a sequential scale (typically of a single hue). For example, the results presented in Fig. 3c and 3d would be clearer if presented using a monochrome sequential scale. The authors should consider this change here and in several other heatmaps presented in figures and supplementary figures.

10) There is an apparent contradiction between the results presented in Fig. 4b and 4c for glial

markers in the hippocampus.

Minor points:

11) In page 7, the authors said "we generated two NeuN+ libraries to supplement the two previously published cell-type Hi-C chromosomal contacts maps from adult CamK-Cre+,Setdb12lox/2lox with littermate controls", which publication are they referring to? Please add the reference.

12) In page 6, the description is misleading: "We ran ...of sorted NeuN+ neuronal and, separately, NeuN- non-neuronal nuclei collected from adult cerebral cortex of three C57Bl6/J mice and two additional animals of mixed genetic background, generating for each of the 8 cell-type specific samples." The description suggest that 10 samples (2x[3+2]) were generated. Please rephrase the description to clearly indicate that only 8 samples were generated.

13) Also, in page 6, what means "animals of mixed (predominantly C57Bl6/J) genetic background". Why did the authors choose to use animals of undetermined genetic background? Is this information used in the analyses?

14) In page 8, what do the authors mean by "overlying cortex"?

15) Fig. 1c: please describe the meaning of the dashed line box in the legend. Similar boxes could be also added in Fig. 1d, 1e and 1h.

16) Fig. 1g: could the authors introduce a color code for the lines to highlight the differences between NeuN+ and NeuN- chromatin?

17) Fig. 1g: I had problem interpreting the small Pearson correlation heatmap presented at the right of this panel. Is the notation correct? A better description of the figure could clarify the situation.

18) Fig. 2d: please describe the color code used in the box and whiskers graph.

19) Fig. 2e: please provide a better description of the figure.

20) Fig. 3b: please provide a legend for oval size (also for circle size in Fig. S12).

21) Fig. 3e and 3f: please revise the contrast of the figure. The bars are difficult to see after printing the figure.

22) Fig. 3g presents an ambitious circos plot, but the graph has such amount of information that I am concerned about its utility. Maybe the authors could introduce some boxes or arrows to assist the reader in detecting the relevant information presented in the plot.

23) Fig. 5c, the text "IAP-gag" should be labeled in pink rather than in red. The authors could also indicate that Fig. 5c shows the detection of the mRNA and Fig. 5c the detection of protein.

Dear Reviewers,

We sincerely appreciate all the thoughtful feedback we received on our submission. We are pleased to report that, over the past few months, we have been able to address all the provided comments and suggestions. As part of these revisions, we have generated additional figures, supplemental figures, and supplemental tables to reflect our new analyses, indicated here:

Newly added panels in the main figures:

- Figure 3 *Panels c) and d): Additional PacBio long read sequencing mapping retrotransposition events in murine germ cells (round spermatids)*
- Figure 4 *Panels a) and b): Microscopy images from cerebral cortex, striatum and hippocampus of Setdb1 mutant and wildtype mice*

Newly added supplementary figures:

- Figure S3 *Correspondence of NeuN+ subcompartment loci across replicates*
- Figure S4 *Subcompartment connectivity across independent studies reveals consistent patterning in adult forebrain neurons.*
- Figure S7 *Hi-C trans profiles in NeuN+ and NeuN-*
- Figure S10 *Strain- and sex-specific heatmap of Hi-C subcompartment interactions*
- Figure S11 *PacBio SMRT long-read sequencing of IAPEzi de novo integration sites in adult mouse cortex and round spermatids* (updated to include additional analyses)*
- Figure S12 *IAP gag DNA FISH in NeuN+ and NeuN- nuclei*
- Figure S14 *Histone methylation at Setdb1-sensitive genes.*
- Figure S17 *NeuN+ and NeuN- H3K9me3 (KO/WT) for ERV classes (I, II, and III)*
- Figure S18 *NeuN+ and NeuN- H3K9me3 (KO/WT) for non-ERV repeat categories*

Newly added supplementary tables:

- Table S4 *Neuronal subcompartment genomic coordinates (GRCm38/mm10) [Espeso-Gil, et. al.]*
- Table S5 *Neuronal subcompartment genomic coordinates (GRCm38/mm10) [Fernandez-Albert, et. al.]*
- Table S6 *Neuronal subcompartment genomic coordinates (GRCm38/mm10) [Bonev, et. al.]*
- Table S7 *ESC subcompartment genomic coordinates (GRCm38/mm10) [Bonev et. al.]*
- Table S22 *H3K9me3 NeuN+ (KO/WT)*
- Table S23 *H3K9me3 NeuN- (KO/WT)*

Our responses to the rest of the feedback follows.

Sincerely,

Sandhya Chandrasekaran (Lead Author)
Schahram Akbarian (Corresponding Author)

Reviewer #1:

Introduction: it consists of a single paragraph that mostly summarizes the results of the manuscript itself. I strongly recommend that the authors add more substance to this section of the manuscript, with the goal of making their study more accessible to a larger audience. Specifically, it would be quite helpful to add some sentences about the genomic technology utilized in the study, which can go immediately after the sentence starting with "However, the relationship..." (pg. 3, line 8). At minimum, Hi-C needs to be described as the state-of-the-art technology to evaluate chromosome conformation capture (3C). The authors need to cite the seminal paper by Lieberman-Aiden et al (2009, Science 326:289-293).

Response: We appreciate this comment and now we cite the paper introducing Hi-C we write '*Here, we apply Hi-C, an established method for genome-wide chromosomal conformation mapping including the spatial segregation of open ('A-compartment') and closed ('B-compartment') chromatin (citing Lieberman-Aiden, et al 2009)*'.

also, Introduction: a few sentences describing previous studies on ERVs in the context of neuroscience, and particularly neuroinflammation, would be a useful addition. Here, I suggest they move a few of their supplementary references to the main text, such as Sharif et al. (2013, Philos Trans R Soc Lond B Biol Sci 368:20110340) and Sankowski et al. (2019, PNAS 116:25982-25990). They may also want to add the reviews by Romer (2021, Front Neurosci 15:648629) and Hantak et al. (2021, Trends Neurosci, DOI:<https://doi.org/10.1016/j.tins.2020.12.003>).

Response: We appreciate this comment and added all of these references as recommended by the Reviewer, into the Introduction and when indicated, in the Discussion section. To comply with journal-mandated formatting requirements and word limits, we tried to be as concise as possible. We wrote into the revised introduction "*Specifically, we identify a neuronal megadomain subtype for which species-specific interaction frequencies track the dramatic reconfiguration of the retrotransposon landscape in Mus musculus-derived inbred lines, primarily due to ongoing germline expansions of ERV Endogenous Retrovirus, a group of retroelements regulated in highly tissue-specific manner (citing Sharif, et al 2013), with neuroinflammatory and neurodegenerative potential (citing Romer), and detrimental effects on cognition upon de-repression (citing Sankowski, et al 2019)*" ...

And we added to the Discussion "*Previous studies linked a small subset of these newly inserted ERV sequences to strain-specific differences in gene expression, pathogenic mutations and neomorphisms (citing Mager, et al 2015; Duhl, et al 1994), thereby providing 'seeds' for ongoing phylogenesis in line with the repurposing of even more ancient Gag and Env sequences into the Arc early response and synaptic plasticity gene (Hantak, et al 2021) and the placental Syncytins (Lavialle, et al 2013) ...*"

Statistical tests: the methods section lacks a separate subsection about statistics. This omission needs to be corrected. Moreover, there are numerous p values within the Results section but very few indicate which test was used. An example of good practice is on pg.4, line 20: " $p=3.36 \times 10^{-41}$, Fisher's 2x2 exact test". The authors need to use this format across the whole manuscript.

Response: We appreciate this comment and now, within the results section, provide information on the specific testing used to generate each p-value information. Furthermore, at the end of the Methods section, we have added a new paragraph, listing all the statistical methods applied, and specify for each figure panel, the specific statistics method applied.

Fig.1d: I suggest that the panels are placed horizontally so that the NeuN+ data is at the left and the NeuN- data is at the right.

Fig.1e: I recommend they place the schematic at the left, the NeuN+ data at the middle, and the NeuN- data at the right; making sure that the A₁, A₂, B₁, and B₂ labels are written explicitly for each dataset. This layout avoids the somewhat incorrect repetition of the schematic in 1e.

Fig.1g: I suggest the color scale for Pearson's correlation is modified, so 0 is represented in blue and 1 by red. This color scale is coherent with the rest of the manuscript. Parenthetically, the color scheme going from dark blue to light blue led this reviewer to think that all the values were near zero. One needs careful examination to realize that the values are actually representing a larger range.

Fig.1g: it is unclear what the ($p < 10E-50$) refers to (inside the figure or in the figure legend). What samples were being compared? Which test was used in this case? I recommend the authors do not add the number in the figure itself, but that they add the information of what test was run in the figure legend. Indeed, they need to make sure that the statistical test is stated for every comparison in the manuscript.

Fig.1h: the graphs showing the Hi-C trans interactions need a Y scale, similar to the scale in Fig.1f. I think a single Y scale at the very left would be sufficient.

Fig.2e: it is unclear why each red arrowhead needs letters, as they do not add extra information.

Response: We appreciate the reviewer's thoughtful suggestions on improving figure clarity and have addressed all the aforementioned minor figure alterations in this submission. The letters and arrowheads in Figure 2E correspond to full-length de novo insertions labeled in Supplementary Table S13 to reference specific coordinates shared among individual samples.

Fig.4a: the GFAP panels are underwhelming. Based on the Western blot information in Fig.4b, one would expect that the GFAP staining is significantly increased in the KO cortex. With a generous eye, one can almost see more blue staining in the deep cortical layers, but I think these panels need to be improved. The authors also need to label the cortical layers (as they do in Fig.5b and 5c).

Fig.4b: it would be great if the authors can show GFAP-stained tissue in addition to the Western blot. I understand they already show data for cortical astrocytes in 4a (with the caveats I just mentioned), but a striatal section (which would apparently show stronger staining in the KO) and a hippocampal section (in which both are similar) would be a nice addition and quite useful to astrocyte aficionados.

Response: We appreciate this comment and have, as recommended by the Reviewer, revised Figure 4a,b (now Figure 5A, 5B), with higher quality histological images from multiple brain regions including cerebral cortex, striatum and hippocampus. The first half of the Figure 5 legend was revised as follows: "(a) Representative images show immunofluorescent signal for GFAP (green) and NeuN (red) in (top) whole brain coronal section (top) and (bottom) hippocampal formation, from wildtype (WT) and *Setdb1* CK-cKO (KO) mice. Gridded rectangles correspond to higher magnification images of cortex in panel b. Single and double asterisks correspond to higher magnification images of hippocampus (a) and striatum (b), respectively. Scale bar = 1mm (a, upper panel); 500um (a, lower panel); 100um (b) (b) (left) cerebral cortex with cortical layers 1-6 as indicated and (right) striatum from section shown in a)." We write in the Results section, "Strikingly, the cerebral cortex and striatal areas of *CamK-Cre⁺,Setdb1^{2lox/2lox}* conditional mutant mice were affected by gliosis and exhibited upregulation of the astrocytic marker, glial fibrillary acid protein (GFAP) (Figure 5A, 5B). Similarly, Iba1 immunostaining marker revealed proliferative and reactive microglia in mutant hippocampus, although labeling of the cortex overlying the hippocampus was not significantly different from control (Figure 5A, 5C, 5D)."

Reviewer #2:

...the authors should substantially simplify their figures, most of which are unnecessarily busy...

Response: We appreciate this reviewer's suggestion to simplify our figures. We have addressed this by reformatting the panels within Figure 1 for better flow, along with appropriate labels. We have also split up Figure 2 from our prior submission into two figures – one which details important points related to our SPRET/EiJ analyses, and one detailing our PacBio studies. Additionally, all figures have been reviewed and edited with appropriate labels and boxes to help facilitate interpretation.

Simple compartment A/B calling is rather stable reflecting euchromatin/heterochromatin, but further classification at subcompartment level is much more variable between experiments. Most importantly, the subcompartments are only computational separation of the genome only based on Hi-C data. There is not a general consensus what the A₁/A₂/B₁/B₂ represent in biology molecularly/chemically besides some non-decisive correlation with various histone marks. Therefore starting this paper by comparing "B₂" in NeuN+ cells to the "B₂" in NeuN- cells is meaningless and an over-interpretation of the Hi-C data; the "B₂" in two cell types are completely different things. This is evident from Fig. 1, there are 104M of B₂-NeuN+ but 396M of B₂-NeuN-, the B₂-NeuN- are actually enriched with some euchromatin features including CTCF and H₃K₄me₁.

Response: We appreciate this comment and in response have conducted additional Hi-C analyses using three independently generated neuronal Hi-C maps previously published by others in the field. We are pleased to report that the neuron-specific compartment organization as we had originally described is highly reproducible, which strongly suggest that this cell type specific higher order chromatin organization is (to use a phrase used by the Reviewer) not meaningless. These additional findings are reported in the first two paragraphs of the results section as follows:

"Interestingly, for subcompartments A₁, A₂ and B₁, >60% of loci in NeuN⁺ and NeuN⁻ nuclei matched (Figure 1B). However, within B₂, the fractional concordance between neuronal and non-neuronal chromatin dropped to <20% (Figure 1B), suggesting that in NeuN⁺, B₂^{NeuN⁺} interact in a unique manner as compared to NeuN⁻.

"Next, to test whether this neuron-specific subcompartment organization is reproducible across independently generated Hi-C datasets, we applied our k-based subcompartment mapping to three published neuron-specific Hi-C data sources from adult mouse cerebral cortex and hippocampus (ref), and from neuronal culture differentiated in vitro (ref) (Tables S4-S6). In fact, B₂^{NeuN⁺} loci reproducibly clustered together specifically in 3/3 of these previously published neuronal Hi-C datasets (Figure S4). In striking contrast, Hi-C maps generated from mouse embryonic stem cells (ESC)²⁴ completely lacked k-clustering of B₂^{NeuN⁺} sequences (Table S7). Furthermore, trans interaction frequencies among B₂^{NeuN⁺} were consistently high in each dataset generated from mature neurons from adult forebrain, while their surrounding non-neuronal cells (B₂^{NeuN⁻}), like ESC or immature neurons, showed markedly weaker B₂ interactions, in contrast to otherwise similar interaction profiles for the remaining subcompartments (Figure 1C, Figure S4). These findings, taken together, confirm that the neuronal 3D genome includes a cell-type specific B compartment subtype with high inter-chromosomal contact frequencies in mature brain."

If we only focus on B₂ in NeuN+, how consistent is B₂ calling among all biological replicates (2F/2M mice)? The authors should provide a direct comparison instead of using Fig.1g.

Response: We appreciate the Reviewer's suggestion to assess consistency of B₂ call among biological replicates. We have included a comparison of subcompartments separately called within female and male replicates using our Hi-C data in Supplementary Figure S₃. To generate high resolution Hi-C maps for interpretation, we compressed samples by sex (4 biological replicates into 2 Hi-C files, so the provided comparison in the figure is effectively for n=2 vs. n=2 and demonstrates 93-96% overlap of loci within each of the four subcompartments. We have incorporated this finding into the text as follows: *"We identified 4 large chromosomal subcompartments in each population (Figures S₁, S₂), reproducible across replicates (Figure S₃),..."*

3. The authors claim from Fig 1g about trans contacts that: (i) the 4 mice are similar; (ii) there are differences between NeuN+ and NeuN-. However, the most recognizable "hotspots" of the trans-contacts shared between 4 mice NeuN+ are also hotspots in NeuN- cells, they do not look to be cell type specific.

4. Fig. 1h shows the best recognizable hotspots from Fig. 1g but try to show they are at different locations on NeuN+ and NeuN-. For example, the hotspot on chr₄ in NeuN+ and NeuN- seems to be the same in Fig. 1g, but Fig. 1h shows that the chr₄ trans-contact hotspots are different between NeuN+ and NeuN- cells? However, if hotspots are truly cell type specific, why do they appear on the same chromosomes? This is a paradox.

Response: We very much appreciate these comments. As the Reviewer's confusion is likely due to the limited spatial resolution of the Hi-C whole genome circoplots in Figure 1g, we now included a 'circoplot zoom-in' for the

50 megabases of chromosome 4 remarked upon by the Reviewer. That zoom in shows that the trans-chromosomal contacts, within the 50Mb window, sharply diverge between NeuN+ neurons and the NeuN- non-neurons. We updated the figure legend accordingly. Furthermore, we added in Figure 1h y-axis scales, and, related, added Supplementary Figure S7 with standardized percentiles on the y-axis, to better document, for multiple chromosomes, the quantification for the sharply diverging peak profiles between neuronal and non-neuronal nuclei. Importantly, as shown in Figure 1h, both cell types show trans-chromosomal contacts within these daily broad (50Mb) chromosomal windows, but at sharply different positions within the 50Mb. We added the following phrase to the results section “with sharply divergent, highly cell-type specific trans-contact profiles as illustrated in the 50Mb wide browser windows from several chromosomes (Figure 1H).”

Relevant to #3 and #4. mm10 is not perfect at the repeats. Calling trans-contact from Hi-C can be wrong if there are translocation/copy number variants/structure variants. Therefore, interpreting trans-interactions at transposons is especially risky. How can the authors exclude the possibility that the observed trans-contacts domains are not due to those reasons? Unannotated transposition may actually explain the paradox mentioned in #4. For example, if there is active viral insertion in NeuN+ cells, that will explain the gain of trans-contacts (due to more copies); some copies may gain H3K9me3 and lead to increased the H3K9me3 ChIP-seq signal. These can be mis-interpreted as “subcompartment”.

Response: We appreciate this comment and in response, conducted additional PacBio long-read assays to capture IAP retrotranspositions in a non-brain cell type (male germ cells) and in addition, we directly compare IAP retrotranspositions events between neuronal and non-neuronal brain cells. We now added the following text to the last paragraph of the result section, subchapter ‘*Neuron-specific megadomain organization tracks recent expansions in ERV genomic landscapes*’: ‘Importantly, the number of IAPEzi genome integration sites in cortical NeuN+ neurons was only minimally different from the corresponding counts in the non-neuronal (NeuN-) nuclei (Figure 3C, Table S13). To examine this further, we conducted additional SMRT-seq of IAPEzi integration sites from two samples of male germ cells, prepared as post-meiotic round spermatid cultures. These experiments revealed an additional set of 439-488 proviral (non-solo-LTR) IAPEzi integration sites not found in mm10, including 47-59 sites/culture harboring full-length IAPEzi. Strikingly, de novo integration sites from these germ cell samples, compared to IAPEzi SMRT-seq integration sites identified in brain cells, showed a very similar type of enrichment for sequences assembling as B₂^{NeuN+} (Figures 3C, 3D). These findings, taken together, effectively rule out neuron-specific retrotransposition as a driver for the observed cell type-specific chromosomal interactions in neuronal nuclei, including B₂^{NeuN+}.’

I therefore think the authors cannot claim “subcompartment” solely based on Hi-C. Independent experiments (eg. FISH) is necessary.

Response: We thank the Reviewer for this interesting suggestion. We are pleased to report that we succeeded in our efforts to stain adult mouse brain sections with NeuN immunofluorescence (to mark neuronal nuclei) and DNA-FISH with a probe recognizing a mobile IAP element. We included our staining technique in the methods section, added Figure S12 to the Supplementary Figure set and we revised the Results sections (subchapter ‘*Neuronal megadomain organization...*’) as follows:

‘Visual inspection of histological sections from mutant and wildtype adult cerebral cortex processed by DNA-FISH with a 300bp IAPEzi gag probe and counterstained with NeuN immunofluorescence and the nucleophilic dye, DAPI, revealed patch-like accumulations of ERV/IAP-containing genomic sites, with some the largest densities most prominently visible in wildtype NeuN+ nuclei (Figure S12), consistent the high physical interaction frequencies of ERV-rich chromosomal loci in Hi-C. Therefore, in order to test whether *Setdb1* could exert a broad regulatory footprint of *Setdb1* on ERV-rich megadomains, including compartmental organization and chromosomal connectivity of neuronal B₂^{NeuN+}, we generated two NeuN+ libraries to supplement the two previously published cell-type Hi-C chromosomal contacts maps from adult *CamK-Cre+*, *Setdb1*^{2lox/2lox} with littermate controls (*CamK-Cre-*, *Setdb1*^{2lox/2lox}) (N=4/group (2M/2F))’

8. I have no problem with Fig. 4-5 but they do not support the subcompartment.

Response: We appreciate this comment and now write at the beginning of the result subchapter '*Gliosis and genomic activation of microglia associated with IAP invasion of neuronal somata and processes*': "We next asked whether the disintegration of B₂NeuN⁺ chromosomal connectivity, including ERV2 de-repression, in our Setdb1-deficient mice could affect neuronal health and function. To this end, analysis of the RNA-seq from Setdb1 mutant as compared to control cortex revealed that the top 10 ranking gene ontology.....". This subchapter discusses Figure 4 and Figure 5 (now Figures 5 and 6), and by writing the above sentence before we discuss Figure 5 and Figure 6, we therefore ensure that the reader will not confuse Figure 5 and Figure 6 as discussion on the subcompartment structure but rather as a discussion of the phenotype associated with the dysregulation of the subcompartments.

9. The discussion about phase separation is speculation. The authors do not have experiments showing that HP1 or heterochromatin is organizing subcompartment via phase separation.

Response: We appreciate this important comment and in response have modified the discussion as follows, in order to put our results in proper context to the state of knowledge in the field:

"Importantly, despite widespread rewiring and disruption of compartment-specific chromosomal patterning, Setdb1-deficient neurons only show few changes in the topologically-associating domain (TAD) landscape, with the notable exception of the clustered Protocadherin locus and a few additional genomic sites (ref). Albeit speculative at this point, this apparent phenotypic dichotomy in the Setdb1 deficient mice, with widespread compartment alterations but much more limited changes in TAD landscapes, could reflect non-overlapping regulatory mechanisms governing these two types of chromosomal conformations. For example, phase separation, as a molecular force, shapes compartments, while actively driven loop extrusions present as TADs in ensemble Hi-C data (ref). Thus, the strong interconnectivity of B₂^{NeuN⁺} megadomains, and their regulatory effects on the surrounding A₂^{NeuN⁺} subcompartment, could depend on 'bridges' of heterochromatin-associated protein 1 (HP1) bound to H3K9me2/3-tagged nucleosomes (ref) and additional phase separation-promoting mechanisms (ref). To this end, it is notable that, according to the present study, genes showing upregulated expression after neuronal Setdb1 ablation are primarily affected by loss of A₂^{NeuN⁺}-B₂^{NeuN⁺} chromosomal contacts but do not show evidence for direct transcriptional regulation by Setdb1 at the site of the gene. These findings, taken together, suggest that in mature cortical neurons, the mechanisms of transcriptional inhibition include balanced interactions between open (A₂^{NeuN⁺}) and repressive (B₂^{NeuN⁺}) megadomains."

Reviewer #3:

I may be overinterpreting the main findings, but I believe that the study suggests that neurons have evolved a megadomain architecture that play a defensive role (referred by the authors as "adaptive") against ERV2 intragenomic parasites. When this architecture is disrupted by the loss of the Setdb1 safeguard, the de-repressed domain unleashes ERV proviruses with strong tropism within mature neurons, leading to neuropathology. Although this result is presented in the frame of the characterization of a mouse strain (Setdb1 KO) that does not model any particular disorder, chromatin de-repression has been proposed as a mechanism of etiopathology in various rare neurodevelopmental disorders, and the erosion of epigenetic profiles and neuroinflammation are features of numerous neurological conditions and the aged brain. The authors could highlight the relevance of their findings by investigating if ERV2 derived transcripts are upregulated in transcriptome datasets for some of these conditions, which could further underscore the defensive role of Setdb1 (and likely other epigenetic enzymes).

Response: We appreciate this comment and added additional discussion and references into the Discussion section, to cover the emerging links between HERV-Ks (the human ERVs most closely related to the ERV2s in the mouse) and neurological disease. To comply with journal-mandated formatting requirements and word limits, we tried to be as concise as possible. We write:

"The findings are also of interest from the viewpoint of the potential neurotoxic effects of ERV2-like transcripts and proteins in human and invertebrate neurons, including potential links to tau- and TDP-43 associated neurodegenerative disease (ref), and recent reports on increased HERV activation in the adolescent non-human primate brain exposed to maternal immune activation during prenatal development (ref). Interestingly, a subset HERVs and other repeat elements are reportedly overexpressed in Alzheimer's brain (ref). Furthermore, Alzheimer-related neurodegenerative phenotypes are encountered in mouse models exposed to specific types of HERV-K RNAs (ref). Moreover, HERV-Ks—which, like the murine ERV2s, are members of the betaretrovirus-like supergroup of LTR retrotransposons— have been linked to amyotrophic lateral sclerosis (ALS) or motor neuron disease (ref) (but see also (ref)), including some cases infected with the Human Immunodeficiency Virus (HIV), an exogenous LTR retrovirus (ref). In addition, the genetic risk architecture of ALS shows a modest association with genomic loci harboring HERV insertions (ref). Interestingly, we observed that genomic loci harboring HERV-Ks are significantly enriched in trans chromosomal Hi-C contacts in human neurons ($p=1.42 \times 10^{-16}$, Fisher's 2x2 exact testing), an effect driven primarily by sequences with strong synteny with murine B₂ megadomains (Figure S19, Figure S20). These findings, taken together, strongly suggest that compartment-specific enrichment of ERV2/HERV-K sequences occurs in parallel across different mammalian lineages, pointing to a type of heterochromatic organization highly adaptive to species- and strain-specific reconfiguration of the ERV retrotransposon landscape, thereby maintaining a defensive shield to protect neurons from the detrimental effects of LTR retrotransposons."

Recent studies have generated Hi-C datasets from mouse brain with more resolution than the data presented here. The authors should confirm that their conclusions regarding the high incidence of trans interaction in the B₂ compartment, etc, are reproduced in these other Hi-C studies. They may also take advantage of the data generated in different developmental stages (from stem cells to mature neurons; see, for example, the studies by Bonev et al. 2017 and Fernandez-Albert et al, 2019) to further explore the cell-type-specific differences in the B₂ compartment described here].

Response: We thank the Reviewer for this great suggestion. We now analyzed independently generated Hi-C data from neurons and stem cells, as suggested by the Reviewer, including the Bonev et al., Fernandez-Albert et al and Espeso-Gil datasets. The new data are presented in the newly added Figure S4 and Tables S4-S7. We observe that the neuron-specific B₂ compartment is highly reproducible in these independent datasets. We describe the additional analyses in the results section:

"Next, to test whether this neuron-specific subcompartment organization is reproducible across independently generated Hi-C datasets, we applied our k-based subcompartment mapping to three published neuron-specific Hi-C data sources from adult mouse cerebral cortex and hippocampus (citing Espeso-Gil, et al 2021; Fernandez-Albert et al 2019), and from neuronal culture differentiated *in vitro* (citing Bonev, et al 2017) (Tables S4-S6). In fact, B₂^{NeuN+} loci reproducibly clustered together specifically in 3/3 of these previously published neuronal Hi-C datasets (Figure S4). In striking contrast, Hi-C maps generated from mouse embryonic stem cells (ESC) (ref) completely lacked k-clustering of B₂^{NeuN+} sequences (Table S7). Furthermore, *trans* interaction frequencies among B₂^{NeuN+} were consistently high in each dataset generated from mature neurons from adult forebrain, while their surrounding non-neuronal cells (B₂^{NeuN-}), like ESC or immature neurons, showed markedly weaker B₂ interactions, in contrast to otherwise similar interaction profiles for the remaining subcompartments (Figure 1C, Figure S4). **These findings, taken together, confirm that the neuronal 3D genome includes a cell-type specific B compartment subtype with high inter-chromosomal contact frequencies in mature brain."**

Given the cellular complexity of the adult cerebral cortex, both the NeuN+ and the NeuN- populations contain a mix of cell types. However, the NeuN- population is expected to be more heterogeneous than the NeuN+ population, and likely contain more divergent cell types. Can this heterogeneity explain the weaker contacts observed in the NeuN- population? Related to the previous point, the authors could take advantage of published Hi-C data for specific cell types to explore the cell type specificity of B₂ presented in the first part of the manuscript.

Response: We thank the Reviewer for this remark. We now have, as described in detail in the preceding remark, analyzed, in addition to our Hi-C data from non-neuronal (NeuN-) nuclei population in adult brain, also analyzed the Hi-C data from Bonev et al. 2017 mouse embryonic stem cells and young, immature neurons. Results,

summarized in the preceding point-by-point response, show that stem cells lack the neuron-specific B2 subcompartment while the remaining A and B subcompartments are present. Immature neurons, like adult neurons, include the B2 subcompartment but the inter-chromosomal contacts (B2 to B2) are much weaker than in mature neurons.

The authors could demonstrate that ERV2 transposable elements are differently H3K9me3-enriched in the NeuN+ and the NeuN- populations by ChIP assay. This would further support a direct role of H3K9me3 (Setdb1 substrate) in neuron-specific B2 megadomain interactions. In the subsequent analysis of Setdb1 KO the authors only analyze the changes in NeuN+ population and they do not analyze the consequences of Setdb1 depletion in the NeuN- population.

Response: We appreciate this comment and conducted, as requested by the Reviewer, additional analyses in the NeuN- population in n=3 mutant and n=3 control mice. We describe these additional data, in the last paragraph of the results subchapter 'Gliosis and genomic activation of microglia associated with IAP invasion of neuronal somata and processes' as follows:

"We observed that in our CamK-Cre⁺,Setdb1^{2lox/2lox} mice with neuronal Setdb1 depleted, there was increased transcription specifically of ERV2s, in conjunction with significantly decreased H3K9me3 levels at ERV2 sequences in neuronal chromatin (p=0.02366, Linear regression) (Figure 6A, Figure S17). In contrast, non-neuronal chromatin from CamK-Cre⁺,Setdb1^{2lox/2lox} mice showed complete preservation of ERV2-bound H3K9me3 (Figure S18, Tables S22, S23). In addition to these RNA-seq based studies, we also observed increased transcription of IAP-gag in cortical neurons by RNA FISH (Figure 6B)."

In the comparative analysis of the two mouse strains (Spret/EiJ and C57/BL6J), the authors do not show the subcompartmentalization between A1, A2, B1 and B2 in cis and trans (similar to what it is presented in Fig. 1C) to verify that the B2 NeuN+ population continues to have a specific compartmentalization in Spret/EiJ background although, the interaction frequency is lower compared to the strain C57/BL6J.

Response: We appreciate this comment and conducted the additional analyses (shown in the newly added Figure S10). We revised the results section as follows:

"Indeed, hotspots of ERV2 repeats (n=101 250kb bins), defined as >99th percentile densities of ERV2 in the mm10 reference genome, interacted less frequently in SPRET/EiJ versus C57/BL6J NeuN⁺ (p=2.25x10⁻¹⁸, Wilcoxon sum-rank testing); in comparison, interaction frequencies at these loci minimally differed in NeuN⁻ nuclei across the two strains (p=0.001, Wilcoxon sum-rank testing) (Figure 2C), supporting a role for ERV2 densities in shaping the 3D NeuN⁺ genome. Furthermore, overall intra- and inter-chromosomal connectivity across the 4 neuronal subcompartments showed minimal changes between SPRET/EiJ and C57/BL6J (Figure S10). We conclude that B₂^{NeuN⁺} megadomain conformations in mature cortical neurons show 'dosage-sensitivity' for phylogenetically young ERV2 sequences."

The transcriptional analysis of the Setdb1 KO mice focused on the changes related to ERV2, but there are also many transcriptional changes in the A2 megadomain. Are DEGs in the A2 megadomain also associated with H3K9me3 enrichment? It would be interesting to show a more detailed dissection of the transcriptional results, exploring the type of DEGs upregulated in each megadomain.

Response: We appreciate this comment and have added the following information to the results section, in the final paragraph of the subchapter 'Neuronal megadomain reorganization upon ablation of Setdb1 methyltransferase': *"Of note, of the 764 unique upregulated autosomal gene transcripts, only a very small subset 46 genes (6%), of which 36 were located in B₂^{NeuN⁺}, had shown significant NeuN⁺ specific H3K9me3 enrichment at baseline (p < 0.001, diffReps negative binomial testing) (Figure S14). These findings, taken together, strongly suggest that genes with increased expression after Setdb1 ablation are primarily affected by loss of chromosomal interactions with repressive B₂^{NeuN⁺} chromatin but are unlikely to be regulated by Setdb1 at the site of the gene body."*

And we added the following sentence to the second paragraph of the discussion section:

'To this end, it is notable that, according to the present study, genes showing upregulated expression after neuronal Setdb1 ablation are primarily affected by loss of A₂^{NeuN+}-B₂^{NeuN+} chromosomal contacts but do not show evidence for direct transcriptional regulation by Setdb1 at the site of the gene. These findings, taken together, suggest that in mature cortical neurons, the mechanisms of transcriptional inhibition include balanced interactions between open (A₂^{NeuN+}) and repressive (B₂^{NeuN+}) megadomains.'

In page 8, the authors conclude that the analysis of DEGs in Setdb1-KOs "potentially indicates a hypermetabolic state in response to an inflammatory stimulus". Is this a cause or a consequence of the epigenetic alterations reported in these animals? The authors could propose a causal relationship between their various observations in these animals in the Discussion section.

Response: We appreciate this important comment and now have added the following text and references to the discussion section: 'The functional importance of proper heterochromatic organization in neurons is underscored by the massive proliferation of IAP particles following the destruction of B₂^{NeuN+} in susceptible *Kmt1e/Setdb1* mutant neurons. This, in turn, is likely to cause proteotoxic stress, with the two top ranking GO categories in the differential RNA-seq analysis (*Figure S15*), 'Endoplasmic Reticulum (ER) stress response' and 'cytosolic ribosomal protein', reflecting the cell's transcriptional response to excessive translational demand⁴⁰. Furthermore, double-stranded viral IAP RNAs⁴¹ could trigger neuroinflammation, including astrogliosis and microglial activation of immune and interferon response genes (*Figure 5*), followed by a more generalized hypermetabolic response in the inflamed brain (*ref*).'

In general, the authors should revise the figure legends. Some panels are very complex and their succinct descriptions do not facilitate their comprehension.

Response: We appreciate the reviewer's suggestion to revise our legends to facilitate comprehension of our figure panels. All legends now include all statistical tests used for associated panels, with guides to interpretation as needed.

While some heatmaps benefit from using a divergent color scale, such as the one presented in Fig. 3b in which the neutral color is at the middle of the scale. However, other heatmaps work better using a sequential scale (typically of a single hue). For example, the results presented in Fig. 3c and 3d would be clearer if presented using a monochrome sequential scale. The authors should consider this change here and in several other heatmaps presented in figures and supplementary figures.

Response: We appreciate this reviewer's feedback on updating our color scales. Reviewer 1 also had suggestions on altering color legends to preserve consistency across the paper. We have made changes to address both reviewers' heatmap scale suggestions as best as possible, except in cases of conflicting advice.

There is an apparent contradiction between the results presented in Fig. 4b and 4c for glial markers in the hippocampus.

Response: We apologize for the substandard quality of our GFAP immunostainings. We repeated the immunohistochemical staining and now completely revised Figure 5 (previously Figure 4), with panel a) showing whole coronal immunostainings, with zoom-ins of the hippocampal areas, in mutant and control mice, and panel b) shows corresponding zoom-ins from cortex and striatum.

Minor points:

11) In page 7, the authors said "we generated two NeuN+ libraries to supplement the two previously published cell-type Hi-C chromosomal contacts maps from adult CamK-Cre+,Setdb12lox/2lox with littermate controls", which publication are they referring to? Please add the reference.

12) In page 6, the description is misleading: "We ran ...of sorted NeuN+ neuronal and, separately, NeuN- non-neuronal nuclei collected from adult cerebral cortex of three C57Bl6/J mice and two additional animals of mixed genetic background, generating for each of the 8 cell-type specific samples." The description suggest that 10 samples (2x[3+2]) were generated. Please rephrase the description to clearly indicate that only 8 samples were generated.

R: "We ran single molecule PacBio SMRT-sequencing (SMRT-seq) on captured DNA of sorted NeuN+ neuronal and, separately, NeuN- non-neuronal nuclei collected from adult cerebral cortex from two independent mouse colonies, generating 2.64-4.70x10⁶ high fidelity (HiFi, >99.9% accuracy) circular consensus sequences (CCS) for a total of 8 cell-type specific samples (5 NeuN+, 3 NeuN-) passing QC metrics and sent for next-generation sequencing (Table S12, see Methods)."

13) Also, in page 6, what means "animals of mixed (predominantly C57Bl6/J) genetic background". Why did the authors choose to use animals of undetermined genetic background? Is this information used in the analyses?

R: See above; we have collapsed this statement into "from two independent mouse colonies" to minimize confusion.

14) In page 8, what do the authors mean by "overlying cortex"?

R: "cortex overlying the hippocampus"

15) Fig. 1c: please describe the meaning of the dashed line box in the legend. Similar boxes could be also added in Fig. 1d, 1e and 1h.

R: Dashed boxes have been added to Figs 1d and 1e, with descriptions in the legend. Boxes have not been added to 1h to preserve figure clarity, but a guide to interpretation has been included in the respective legend.

16) Fig. 1g: could the authors introduce a color code for the lines to highlight the differences between NeuN+ and NeuN- chromatin?

R: We have highlighted differences between NeuN+ and NeuN- on a Mb-scale in Figure 1H; furthermore, we have incorporated a "zoom-in" schematic to highlight the relationship of the 50Mb windows in Figure 1h to the Circos plots in Figure 1G.

17) Fig. 1g: I had problem interpreting the small Pearson correlation heatmap presented at the right of this panel. Is the notation correct? A better description of the figure could clarify the situation.

18) Fig. 2d: please describe the color code used in the box and whiskers graph.

19) Fig. 2e: please provide a better description of the figure.

20) Fig. 3b: please provide a legend for oval size (also for circle size in Fig. S12).

21) Fig. 3e and 3f: please revise the contrast of the figure. The bars are difficult to see after printing the figure.

22) Fig. 3g presents an ambitious circos plot, but the graph has such amount of information that I am concerned about its utility. Maybe the authors could introduce some boxes or arrows to assist the reader in detecting the relevant information presented in the plot.

R: We have made some figure alterations to better situate to Circos slice in Figure 3G into the overall Circos plot; additionally, we have included a guide to interpretation in the figure legend to highlight relevant information.

23) Fig. 5c, the text "IAP-gag" should be labeled in pink rather than in red. The authors could also indicate that Fig. 5c shows the detection of the mRNA and Fig. 5c the detection of protein.

Response: We thank the reviewer for their suggestions on improving our manuscript and have addressed all comments in the manuscript and figures; specific points have been elaborated above.

REVIEWER COMMENTS

Reviewer #1 (Remarks to the Author):

In their revised manuscript, the authors have satisfactorily addressed all my comments and concerns. In particular, the authors have improved key points in the figures and have expanded the text. In my opinion, the work strongly supports the author's conclusions.

Incidentally, I've noted two very minor points:

typo in line 208: "subset 46 genes" should read "subset of 46 genes"

typo in Fig.S14: "minimally eniched" should read "minimally enriched"

Reviewer #2 (Remarks to the Author):

My previous concerns focused on the analysis of Hi-C sub-compartments. Overall the authors made a lot of efforts improving this manuscript and have addressed most of my concerns. It is good to see consistent B2 subcompartments between biological replicates and in brain tissues. I also appreciate the newly added PacBio data (Figure 3), which addressed my question if active transposition affects their analyses.

I had a major issue in my previous review about the comparison between the B2 subcompartment in NeuN+ against the B2 subcompartment in NeuN- cells. I still think this is not an appropriate comparison. The reason is that although I am now convinced that B2-NeuN+ is real, it is still unclear what B2-NeuN- is. What is common between B2-NeuN+ and B2-NeuN- except they are both named "B2"? It does not make sense to compare B2-NeuN+ to B2-NeuN- without knowing what B2-NeuN- is. In another word, we can rename B2 in NeuN- cells as C1 because they have little in common with the B2 in NeuN+ cells.

I want to make myself clear that although this is a conceptual flaw, I don't think this problem is a deal breaker for the whole paper because the story of this paper does not need to compare B2-NeuN+ to B2-NeuN-. All the authors need to show is that B2-NeuN+ subcompartment does not exist in NeuN- cells. Since the authors already show the preferential cis and trans interactions between B2-NeuN+ regions, they can directly show if these interactions disappear or weaken in NeuN- cells. One possible way to show this can be found in Figure 3 of a recent preprint <https://doi.org/10.1101/2021.04.23.441217>. Notably, this preprint also reports the preferential interactions between H3K9me3 occupied regions but in a different context.

Reviewer #3 (Remarks to the Author):

The revised manuscript is significantly improved. It includes new experiments and analyses and effectively addressed the most relevant criticisms.

Dear Reviewers,

We sincerely appreciate all the thoughtful feedback we received on our re-submission. We are pleased to report that we have been able to address all the provided comments and suggestions. As part of these revisions, we have generated an additional supplemental figure, indicated here:

Newly added supplementary figure:

Figure S8 *Trans interactions occur less frequently among B_2^{NeuN+} loci in NeuN- as compared to NeuN+*

Our responses to reviewer feedback follow on the subsequent pages of this document.

Sincerely,

Sandhya Chandrasekaran (*Lead Author*)

Schahram Akbarian (*Corresponding Author*)

Reviewer #1:

Minor points:

- typo in line 208: "subset 46 genes" should read "subset of 46 genes"
- typo in Fig.S14: "minimally enriched" should read "minimally enriched"

Response: We thank the reviewer for their suggestions and have addressed these points in the manuscript and legends.

Reviewer #2:

It is still unclear what B2-NeuN- is. What is common between B2-NeuN+ and B2-NeuN- except they are both named "B2"? It does not make sense to compare B2-NeuN+ to B2-NeuN- without knowing what B2-NeuN- is. In another word, we can rename B2 in NeuN- cells as C1 because they have little in common with the B2 in NeuN+ cells.

Response: We appreciate the reviewer pointing out this source of confusion. We have included a statement about the subcompartment naming in the manuscript on page 3, "*We assigned identifiers to each (subcompartment) first according to minority (A) or majority (B) fractional concordance with heterochromatic, nuclear lamina-associated domain (LAD) sequences, and then numbered based on decreasing size.*" This labeling is the same strategy employed in Rao, et. al. (2014) and is discussed in further detail in our Methods section. Because the labeling of these subcompartments is performed independently for each cell type, there is no *a priori* requirement for similarly labeled subcompartments across cell types to share loci. Despite this, we have found that loci independently categorized as A1, A2, and B1 in NeuN+ share overlap with loci in A1, A2, and B1 in NeuN- (Figure 1B); it is only loci in B2 in NeuN+ (i.e., loci classified within the smaller NeuN+ cluster within the heterochromatic "B" compartment) that share minimal overlap with loci in B2 in NeuN- (i.e., loci classified within the smaller NeuN- cluster within the heterochromatic "B" compartment). In line with this naming system that we have employed through the paper, based on published precedent in the literature, we find that relabeling B2 in NeuN- as C1 would be misleading, as the "B" refers to the compartment, of which there are only two choices (A (euchromatin) vs. B (heterochromatin)). We hope this explanation serves to sufficiently clarify this issue.

The authors need to show is that B2-NeuN+ subcompartment does not exist in NeuN- cells. Since the authors already show the preferential cis and trans interactions between B2-NeuN+ regions, they can directly show if these interactions disappear or weaken in NeuN- cells. One possible way to show this can be found in Figure 3 of a recent preprint <https://doi.org/10.1101/2021.04.23.441217>. Notably, this preprint also reports the preferential interactions between H3K9me3 occupied regions but in a different context.

Response: We appreciate the reviewer's suggestion, and have included a new supplementary figure, S8, that illustrates significantly weakened (ie, less frequently occurring, $p < 10^{-300}$ [Student's t-test, two-sided, paired]) *trans* interactions in NeuN- as compared to NeuN+ when looking specifically at B₂^{NeuN+} loci. In this figure, we have also included a panel, modeled after Figure 3 of the included preprint, to highlighted differences in these interactions in NeuN+ vs. NeuN-. Finally, we have included a statement in the manuscript referring to these findings on page 5, "*Importantly, loci comprising B₂^{NeuN+} interacted significantly more frequently within NeuN+ as compared to NeuN- nuclei (Figure S8).*"

In addition, we thank the Reviewer for this reference and added a sentence in the discussion with the citation: "*Interestingly, long-range megadomain interactions of H3K9me3-tagged chromatin have recently been implicated in human neurodevelopmental disease associated with instability of short tandem repeats {Zhou, 2021}.*"

EVIEWERS' COMMENTS

Reviewer #2 (Remarks to the Author):

I have compared the latest manuscript to the initial submission carefully. I noticed that the authors have already removed the sentences with direct comparisons between B2-Neu+ and B2-Neu-. Therefore I am okay with the languages in the current manuscript. The new Figure S8 confirmed my impression that B2-Neu+ is dissolved in Neu- cells.

To clarify, I certainly did not suggest the authors rename their B2-Neu- to C1. I say this only to make my point that the naming of subcompartments by Rao et al. is arbitrary. In fact, Rao et al named 4 subcompartments (B1-B4) in GM12878 cells. It is totally possible that some subcompartments will change their property or be completely missing in a different cell type. For example, if B2 is missing in a different cell type, B3 may become B2. This is why I insist that it is meaningless to directly compare B2-Neu+ to B2-Neu-, as the authors did in their first submission. However, it is logically okay to describe how the B2-Neu+ loci change their epigenetic property in a different cell type, as what the current version is.